# Emergence of decadal linkage between Western Australian coast and Western–central tropical Pacific

Yuewen Ding [1,2,3], Pengfei Lin [1,2] ✉, Hailong Liu [1,4], Bo Wu[1], Yuanlong Li [5], Lin Chen[6], Lei Zhang[7], Aixue Hu [8], Yiming Wang[1,2], Yiyun Yao[1,2], Bowen Zhao [1,9], Wenrong Bai[1,10] & Weiqing Han [3]

The impact of interbasin linkage on the weather/climate and ecosystems is significantly broader and profounder than that of only appearing in an individual basin. Here, we reveal that a decadal linkage of sea surface temperature (SST) has emerged between western Australian coast and western–central tropical Pacific since 1985, associated with continuous intensification of decadal variabilities (8–16 years). The rapid SST changes in both tropical Indian Ocean and Indo-Pacific warm pool in association to greenhouse gases and volcanoes are emerging factors resulting in enhanced decadal co-variabilities between these two regions since 1985. These SST changes induce enhanced convection variability over the Maritime Continent, leading to stronger easterlies in the western–central tropical Pacific during the warm phase off western Australian coast. The above changes bring about cooling in the western–central tropical Pacific and strengthened Leeuwin Current and anomalous cyclonic wind off western Australian coast, and ultimately resulting in enhanced coupling between these two regions. Our results suggest that enhanced decadal interbasin connections can offer further understanding of decadal changes under future warmer conditions.

Compounded with substantial warming, the variability in sea surface temperature (SST) off western Australian coast[1–3] and in the western–central tropical Pacific[4–6] can greatly affect the local weather, climate, and marine ecosystems[4,7]. Adjacent to these two regions, the tropical Indian Ocean and Indo-Pacific warm pool have experienced the most rapid warming associated with climate change[8–11] and became less influenced by the Interdecadal Pacific Oscillation (IPO)[12,13]. This rapid warming can cause changes in the Walker circulation[14–16] via modulating convective activities and further influence the emergence of interbasin connections within the tropics[17–19]. The interbasin linkage between off western Australian coast and the western–central tropical Pacific, linked by the rapid warming regions (tropical Indian ocean and Indo-Pacific warm pool) and the overlying atmospheric circulation, can

[1]State Key Laboratory of Numerical Modeling for Atmospheric Sciences and Geophysical Fluid Dynamics (LASG), Institute of Atmospheric Physics (IAP), Chinese Academy of Sciences, Beijing 100029, China. [2]College of Earth and Planetary Sciences, University of Chinese Academy of Sciences, Beijing 100049, China. [3]Department of Atmospheric and Oceanic Sciences (ATOC), the University of Colorado (CU), Boulder 80303 CO, USA. [4]Laoshan Laboratory, Qingdao 266237, China. [5]Key Laboratory of Ocean Circulation and Waves, Institute of Oceanology, Chinese Academy of Sciences, Qingdao 266071, China. [6]Key Laboratory of Meteorological Disaster, Ministry of Education (KLME)/Collaborative Innovation Center on Forecast and Evaluation of Meteorological Disasters (CIC-FEMD), Nanjing University of Information Science and Technology, Nanjing 210044, China. [7]State Key Laboratory of Tropical Oceanography, South China Sea Institute of Oceanology, Chinese Academy of Sciences, Guangzhou 510301, China. [8]Climate and Global Dynamics Laboratory, National Center for Atmospheric Research, Boulder 80307 CO, USA. [9]Shanghai Typhoon Institute, China Meteorological Administration, Shanghai 200030, China. [10]Beijing Municipal Climate Center, Beijing Meteorological Bureau, Beijing 100089, China. ✉e-mail: linpf@mail.iap.ac.cn

 

generate broader and profounder climate impacts than appearing only in an individual basin.

The interannual SST variabilities off western Australian coast and in the western–central tropical Pacific[20–22] have received a great deal of attention over the last decade. The interannual variation in SST off western Australian coast, characterized by the Ningaloo Niño, is related to both local and remote processes[3,23,24]. The western–central tropical Pacific (Niño4w) region is a remote area with a strong linear correlation with Ningaloo Niño variability[4,7]. Conversely, the change off western Australian coast can actively affect variability in the Niño4w region, building a two-way interaction on an interannual timescale[4,7]. The interannual SST variability off western Australian coast is also affected by IPO[25–28], such as an amplified Ningaloo Niño amplitude during the negative phase of the IPO[29,30].

Although the interannual SST variabilities both off western Australian coast and in the western–central tropical Pacific have been studied extensively, the decadal SST variabilities in these regions are much less studied, and their causes remain elusive. The emergence of significant decadal SST variability off western Australian coast is not well reconciled yet[1,2,29]. Feng et al.[29] indicated a timing of the late 1990s, whereas ref. [2] suggested a timing of post-1980. Trenery and Han[31] showed enhanced influences of the tropical Pacific winds on sea level and thermocline depth in the tropical south Indian Ocean and western Australian coast since the early 1990s. Furthermore, it is unclear whether the decadal SST variability off western Australian coast is linked to the decadal variability in the western–central tropical Pacific like that on interannual timescale.

Here, we reveal an enhanced decadal SST variability off western Australian coast and an emergence of a decadal linkage with the western–central tropical Pacific over the past four decades based on three sets of observational data. The analysis of two pacemaker experiments related to tropical Pacific SST from two climate models show that negative IPO can partially influence this decadal linkage. Further analysis of the pacemaker experiments related to Indian Ocean SST, two large ensemble simulations from two climate models, and four single forcing large ensembles, we confirmed that the SST changes in the tropical Indian Ocean and Indo-Pacific warm pool owing to volcanic activities can enhance SST decadal variabilities and their linkage between off western Australian coast and in the western–central tropical Pacific in recent decades, leading in an intensification of the easterly wind anomalies in the western–central tropical Pacific. Moreover, the decadal linkage is related to changes in the variability of convection over the Maritime Continent.

## Results

### Enhanced decadal SST variability off western Australia and its linkage to the Pacific

Here, the SST variability off western Australian coast is represented by the Ningaloo Niño index (NNI), which is defined as the area-weighted mean SST anomaly (SSTA) in the wedged area shown in Fig. 1a (see methods). The standard deviations (STDs) of NNI are 0.44 °C (1950–1985) and 0.51 °C (1985–2020). For these two periods, the STDs of the Niño4w index are 0.42 °C and 0.5 °C, respectively. The changes in NNI amplitude (denoted by STD) show that a substantial increase in SST variability off western Australian coast has occurred since the 1980s (Figs. 1b and S1a, e). This increased variability is consistently shown by all observational datasets. Meanwhile, the SST decadal variability before 1950 is less consistent among these observations. After 1980, the dominant periodicity of NNI changes consistently from the interannual timescale to the decadal timescale (Figs. 1b and S1a, e). Wavelet analysis reveals that the dominant NNI period over the past 40 years (1980–2020) is 8–16 years, with statistical significance at the 95% confidence level (1980–2020; Figs. 1d and S1b, f, S2b, e). Moreover, the NNI amplitude on the decadal timescale (0.33 °C) exceeds that on the interannual timescale (0.26 °C). Note that the enhanced variability

since the 1980s is not due to data inconsistency associated with the availability of satellite observations (Fig. S3). In contrast, no evident enhancement in the decadal SST variability during 1980–2020 is found in other areas around the region off western Australian coast (Fig. S4). This confirms the uniqueness of the change in SST variability in regions off western Australian coast since the 1980s.

The SST variability in the western–central tropical Pacific is represented by the Niño4w index (see methods). Studies show that the Niño4w index can be the leading indicator of the NNI change on interannual timescale[4,5]. Approximately 50% of the year-to-year variation in the Ningaloo Niño occurs in conjunction with La Niña[3,22,32], and the Niño4w index is the one most representative of the effect of La Niña on the Ningaloo Niño[4]. Similar to the Ningaloo Niño, the Niño4w index also shows strong decadal variability in recent decades (Figs. 1c, e and S1c, g), and such decadal variability begins to increase in the early 1960s. Its spectral peak becomes strong and statistically significant in the 1970s (Fig. 1e). Since 1985, the Niño4w amplitude on decadal timescale (0.32 °C) is comparable to that on interannual timescale (0.29 °C). The coincidence of the enhanced decadal variability in both the Niño4w and the Ningaloo Niño regions implies a potential linkage between them since the 1980s.

The NNI and the Niño4w index are negatively correlated from their correlation coefficients using 21-year sliding windows (Figs. 1f and S1d, h, S2a, d). This negative correlation coefficient is enhanced around 1960 (~−0.1), becomes statistically significant (−0.5) in the early 1970s, and is maintained after 1985 (−0.75). To further identify the changing characteristics of the NNI and the Niño4w index, we conducted a cross-spectrum analysis (Fig. 1g) for two periods: 1950–1985 and 1985–2020. There is one common significant peak in the coherence spectrum at 2–7 years in both periods. However, after 1985, a second significant coherence peak appears at 8–16 years (Figs. 1g and S5). This suggests that these two indices covary on interannual timescales before 1985 and covary on both interannual and decadal timescales after 1985, implying a potential decadal linkage between these two regions after 1985. Here, we denote "1985" as the "time of emergence" for the NNI-Niño4w linkage on a decadal timescale, defined as when the cross-spectrum of these two indexes is significant at a 95% level, and their correlation coefficient is stably exceeding 95% significance. In the following, we focus on decadal timescales with 8–16 years.

### Enhanced convective center over the Maritime Continent

To gain insight into why the decadal variability in SST enhanced after 1985 off western Australian coast, we examined the changes in air-sea interactions before 1985 (Fig. 2a, c) and after 1985 (Fig. 2b, d) in association with the NNI. The regressions of SSTA onto the decadal NNI in boreal winter (December–January–February, DJF) display similar positive anomaly patterns over the region off western Australian coast for both periods, with some enhancement for the latter period. In contrast, over the Niño4w region, the anomalies are negative. The center of the negative SSTA shifts westward from the central-eastern Pacific near the dateline before 1985 (Fig. 2a; shading; Fig. S6) to the western–central tropical Pacific after 1985 (Fig. 2b; shading; Fig. S6). Note that positive SSTAs appear in the western tropical Pacific west of the Niño4w region after 1985 while they are negligible before 1985. As the center of negative anomaly spans the entire Niño4w region and both the Niño4w index and NNI are strengthened after 1985, it indicates a strengthened connection between the Niño4w index and the NNI on decadal timescale.

Change in surface wind plays an important role in modulating the SST variability in these regions. Associated with the positive SSTA off western Australian coast, significant wind anomalies appeared in both the tropical Pacific and the Indian Oceans during the two periods. However, the wind anomalies after 1985 are substantially stronger than those before 1985 (Fig. 2a, b; vectors; Fig. S6). On one hand, easterly

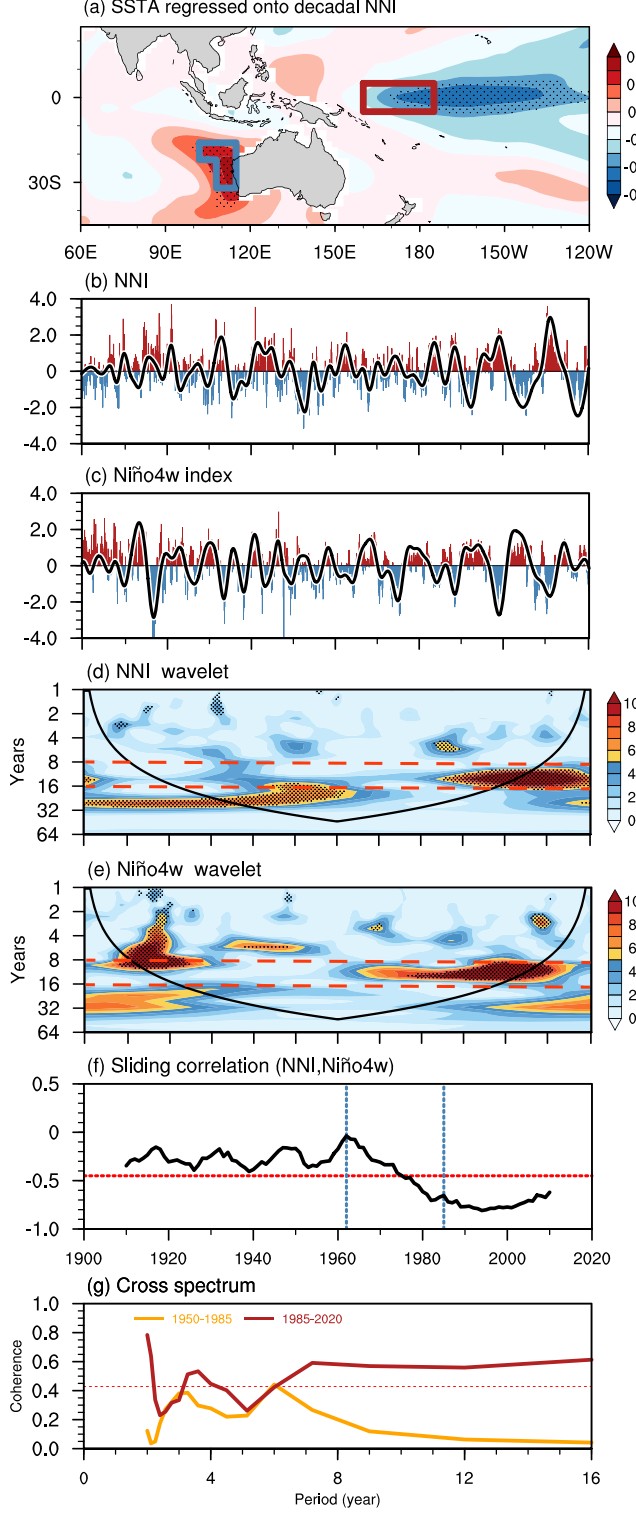

**Fig. 1 | Sea surface temperature (SST) variabilities off western Australian coast and in the Niño4 west (Niño4w) region, their wavelet spectrums, and their relationship. a** Regression of the December–January–February sea surface temperature anomaly (DJF SSTA) onto the Ningaloo Niño index (NNI) based on ERSSTv5. The blue/red box represents the Ningaloo Niño and the Niño4w regions, respectively. Stippled areas denote where the regression results are statistically significant at the 95% confidence level. **b** Normalized NNI (bars) and the decadal component of the NNI from Ensemble empirical mode decomposition (EEMD; black line). **c** same as in **b**, but for the Niño4w index. **d** Wavelet spectrum of the NNI. Black dots indicate areas exceeding the 95% confidence intervals. Areas under the cone-shaped lines are influenced by the edge effect. **e** Wavelet spectrum of the Niño4w index. The red lines mark the periodicity of 8–16 years. **f** The 21-year sliding correlation coefficients between the NNI and the Niño4w index, in which the red line marks where the results are statistically significant at the 95% confidence level. Blue lines represent the turning points (the 1960s, when the NNI–Niño4w connection emerged; 1985, the time of the stable NNI–Niño4w connection). The years in **e** denote the centers of the sliding windows; e.g., 1980 represents the correlation in the 21-year sliding window of 1970–1990. **g** Coherence spectrum of the NNI with the Niño4w index for 1950–1985 and 1985–2020. The red line shows results statistically significant at the 95% confidence level. The blue line marks the periodicity of 8–16 years.

To quantify the underlying physical processes of the decadal SST change in the Ningaloo Niño region, we diagnosed the relative roles of the ocean dynamics, and wind–evaporation–SST (WES) feedback[33,34] (Table 1). The contribution of ocean dynamics to the SSTAs is positive after 1985 (0.15 K), but negative (−0.25 K) before 1985 which means that ocean dynamics damps the SSTAs before 1985, but enhances the SSTAs after 1985. The latent heat flux is nearly unchanged between these two periods (0.32 K before 1985 and 0.33 K after 1985), and thus, the effect of WES feedback is negligible. The damping effect of shortwave radiation on SSTAs significantly enhances in the second period (−0.14 K) than that in the first period (−0.05 K), working against the SSTAs. The contribution of longwave (0.01 K) radiation and sensible heat flux (0.05 K) to SSTAs is generally small, agreeing with previous studies[33,34]. Our results are broadly supported by existing observational studies, although the primary focus of most previous work was on interannual timescale[5].

To further validate the importance of ocean dynamics, we did a composite analysis of the mixed-layer heat budget off western Australian coast (in Ningaloo Niño region, Table 2) during the developing phases of decadal Ningaloo Niño events (see methods). In the developing phase, the Ningaloo Niño region experienced a significant warming tendency, and the warming rate is larger for the post-1985 period compared with the before-1985 period (Table 2). The larger rate can help to develop SSTA. Increased ocean heat advection plays a dominant role in causing the intensified SST variability. As shown in Table 2 and Fig. S7, the largest terms of the heat budget are the surface heat flux and the residue. Here the residue term represents the vertical heat advection due to the upwelling of subsurface water into the mixed layer and all other non-resolved processes. The net effect of the horizontal advection terms is to diverge the heat away from the Ningaloo Niño region, and this effect is larger before 1985 than after 1985. Combined with the residue term, our budget analysis suggests that the reduced heat divergence due to horizontal advection and the reduced upwelling after 1985 led to an enhanced Ningaloo Niño amplitude on the decadal timescale (Fig. S7). Meridional heat advection primarily originates from the western tropical Pacific[29], driving more warm water towards the northwestern Australian coast. With the strengthening of the cyclonic wind anomaly, the resulting eastward geostrophic current anomalies carrying warmer water lead to an overall increase in SST off western Australian coast. Thus, the dynamical process increases the chances of local SSTs rising above the threshold for deep convection, increasing convective cloud[30,35], and reducing solar radiation, which explains the negative effect of solar radiation discussed above.

wind anomalies in the equatorial Pacific can enhance equatorial upwelling cooling and induce westward currents, advecting more cold water from the central tropical Pacific into the western Pacific[25] and causing the strengthened negative SSTA in the Niño4w region. On the other hand, the enhanced anomalous cyclonic wind over the southeast Indian Ocean after 1985 strengthens the northerly wind anomaly. The cyclonic winds drive clockwise geostrophic currents in the Southeast Indian Ocean. The southeastward currents from the tropics, including a more vigorous Leeuwin Current along the coast, lead to stronger heat advection to the Ningaloo Niño region off western Australian coast[21,32].

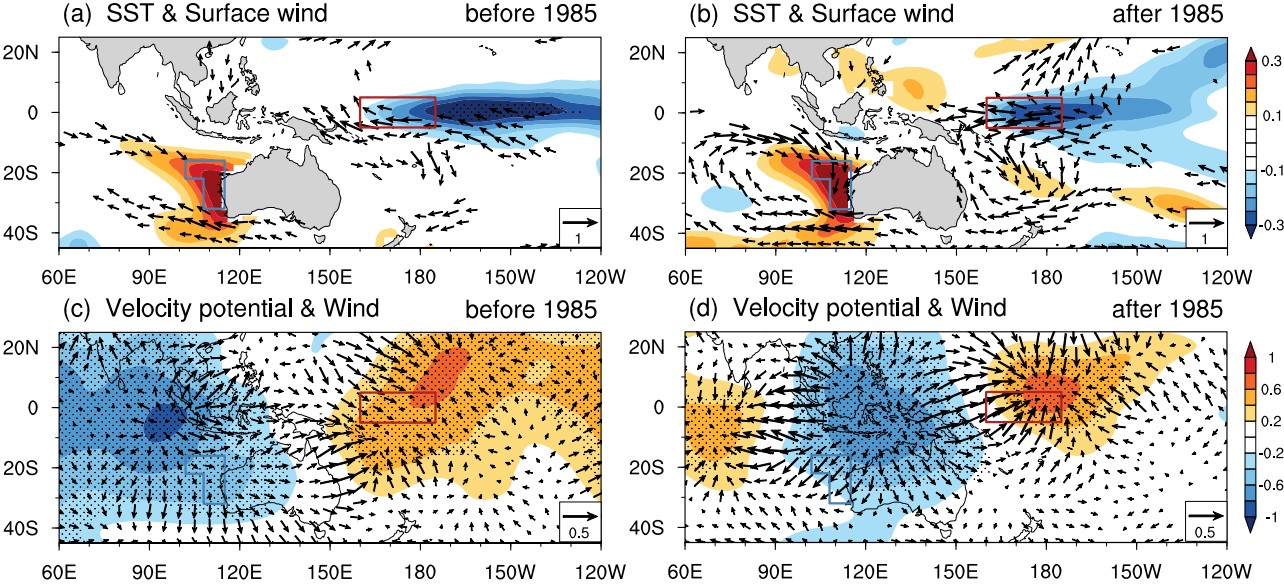

**Fig. 2 | Teleconnections for the decadal sea surface temperature (SST) variability off western Australian coast in December–January–February (DJF).** Regression of the DJF sea surface temperature anomaly (SSTA) onto the DJF decadal Ningaloo Niño index (NNI; shading; unit: K; vectors for wind anomaly; unit: m/s) in **a** 1950–1985 and **b** 1985–2020. The blue/red box represents the region off western Australian coast and the Niño4w region, respectively. Stippled areas are where the regression results are statistically significant at the 95% confidence level. **c**, **d** as in **a** and **b** but for the 200-hPa velocity potential anomaly (shading; unit: 10⁶ m/s; vectors for divergent wind; unit: m/s;). Ningaloo Niño generally peaks in boreal winter; thus, only the changes in boreal winter are presented.

**Table 1 | Contribution of wind–evaporation–SST (WES) feedback and oceanic heat transport to the formation of sea surface temperature (SST) pattern**

|                                                      | Before 1985   | After 1985     |
| ---------------------------------------------------- | ------------- | -------------- |
| SST ($T'$)                                           | 0.12 (0.88)   | 0.4 (0.92)     |
| Latent heat flux to the atmosphere ($Q_{lh}^{a'}$)   | 0.32 (1.21)   | 0.33 (1.29)    |
| Oceanic heat transport ($D'_O$)                      | −0.25 (−0.43) | 0.15 (0.004)   |
| Shortwave radiation ($Q'_{sw}$)                      | −0.05 (0.04)  | −0.14 (−0.48)  |
| Longwave radiation ($Q'_{lw}$)                       | 0.03 (−0.02)  | 0.01 (0.03)    |
| Sensible heat flux ($Q'_{sh}$)                       | 0.07 (0.08)   | 0.05 (0.08)    |

Regression of the December–January–February Sea Surface Temperature Anomaly (DJF SSTA; **T'**, unit: K; Eq. (2)) onto the DJF Ningaloo Niño index (NNI) on decadal timescale and the DJF NNI of original time series without filtering (in brackets) for 1958–1985 and 1985–2015. Also shown are the contributions of latent heat flux to the atmosphere (**$Q_{lh}^{a}$**, unit: K), oceanic heat transport (**$D_O$**, unit: K), shortwave radiation (**$Q'_{sw}$**, unit: K), longwave radiation (**$Q'_{lw}$**, unit: K), and sensible heat flux (**$Q'_{sh}$**, unit: K).

**Table 2 | The average of mixed-layer heat budget terms of Ningaloo Niño region by choosing decadal Ningaloo Niño events (before 1985 and after 1985)**

| Average of decadal Ningaloo Niño events ($10^{-9}$ K/s) | Before 1985 | After 1985 |
| --- | --- | --- |
| $\frac{\partial T'}{\partial t}$ | 1.44 | 1.86 |
| $\frac{Q_{net}'}{\rho_0 C_p h}$ | −12.88 | −9.16 |
| $R'$ | 27.37 | 15.26 |
| $u' \frac{\partial T'}{\partial x}$ | 0.77 | 0.51 |
| $u' \frac{\partial \bar{T}}{\partial x}$ | 0.16 | 2.23 |
| $\bar{u} \frac{\partial T'}{\partial x}$ | −1.51 | 0.64 |
| $v' \frac{\partial T'}{\partial y}$ | 0.79 | −0.74 |
| $v' \frac{\partial \bar{T}}{\partial y}$ | −9.94 | −7.05 |
| $\bar{v} \frac{\partial T'}{\partial y}$ | −6.08 | 1.21 |

The chosen events are December 1970 to December 1972, and January 1981 to August 1983 (before 1985); June 1986 to June 1988, and October 1995 to August 1997 (after 1985), monthly data after 8–16 years band-pass filter is used.

The decadal change in surface wind is associated with changes in both atmospheric convective activity and the circulation pattern. The regression of precipitation onto the decadal part of boreal winter (DJF) NNI shows a positive precipitation anomaly centered around the tropical eastern Indian Ocean (Fig. S8) before 1985, but over the Maritime Continent (Fig. S8) after 1985, implying enhanced ascending (descending) motion over the Maritime Continent associated with Ningaloo Niño (Niña) event after 1985. Correspondingly, the center of the 200-hPa divergence moves to the Maritime Continent, i.e., the pattern of divergence (convergence) over the Indian Ocean (Pacific) before 1985 changes to divergence (convergence) over the Maritime Continent (the Indian Ocean and central-eastern Pacific) after 1985 (Fig. 2c, d; Fig. S6; shading and vectors). The enhanced NN-Niño4w coupling (Fig. 2a, b) is accompanied by enhanced convective activity over the Maritime Continent. This marked enhancement in convective variability in the past four decades (after 1985) is also evident from the singular value decomposition (SVD) results. This is consistent with the spatial pattern of the warming off western Australian coast

during these two periods (Figs. S9–S11). Overall, the enhanced decadal variability of convective activity presented in DJF over the Maritime Continent after 1985 has led to strengthened variability of the Walker circulation and its associated easterly wind anomalies over the tropical Pacific, and to a much lesser extent westerly wind anomalies over the central-east Indian Ocean around 5°S–15°S. Stronger/weaker easterly wind in the Niño4w region will directly cause convergence/divergence over the Maritime Continent region within the warm pool and thus affect convection variability over the Maritime Continent. The decadal changes in the convection center can induce decadal linkage between the western Australian coast and the western–central tropical Pacific.

**Indian Ocean SSTA versus the IPO**

The above analysis confirms that the decadal linkage between the western Australian coast and the western-central tropical Pacific is

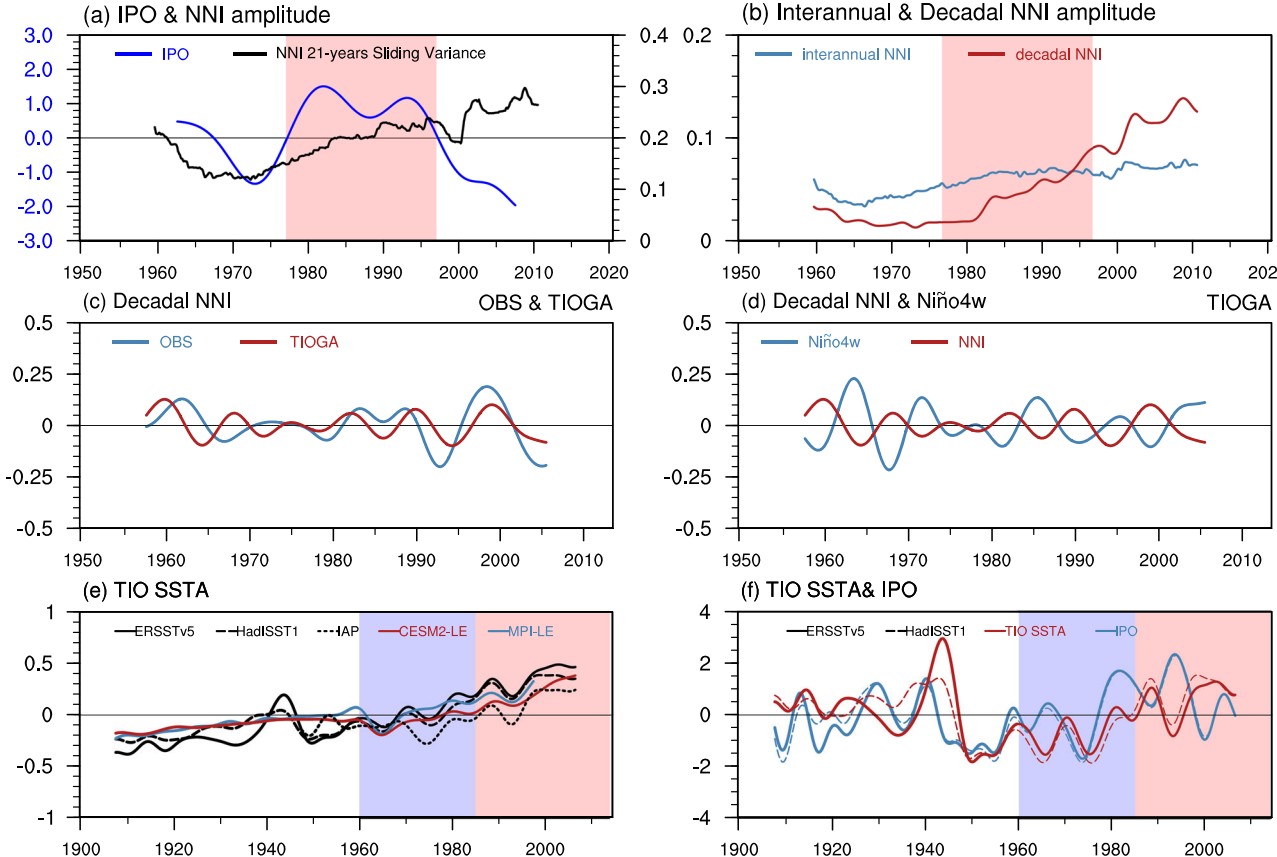

**Fig. 3 | Possible influencing factors (Interdecadal Pacific Oscillation and external forcing) for decadal variabilities of Ningaloo Niño index (NNI). a** Time series of the 21-year sliding variance of the NNI (black line) and the Interdecadal Pacific Oscillation (IPO) index (blue line) based on ERSSTv5, in which the red shading identifies the positive phase of the IPO during 1978–1997. **b** Time series of the 21-year sliding variance of interannual NNI (blue line) and decadal NNI (red line). Red shading identifies the positive phase of the IPO during 1978–1997. **c** The 8–16 years band-pass filtered decadal NNI in ERSSTv5 and tropical Indian Ocean (TIOGA) pacemaker experiments. **d** The 8–16 years band-pass filtered decadal NNI and Niño4w index in TIOGA. **e** The 8 years low-pass filtered tropical Indian Ocean sea surface temperature anomaly (SSTA; 15°N–15°S, 50°–160°E) in observations and the ensemble mean of CESM2-LE, and MPI-LE. Light blue shading identifies the period 1960–1985. Light red shading identifies the period after 1985. **f** The normalized 8-year low-pass filtered (detrend) Tropical Indian Ocean SSTA (red) and IPO index (blue). Solid(dash) represents ERSSTv5 (HadISST).

associated with the enhanced convection variability over the Maritime Continent. The IPO should affect this decadal linkage because the western–central region (especially the Niño4w region) is the equatorial part of the IPO, according to ref. 36. Next, we will not only confirm the effects of the IPO on this interbasin linkage, but also uncover some different factors.

Two sets of pacemaker experiments with two different climate models are employed to investigate the effect of the IPO on the decadal NNI by restoring the simulated SSTAs to those observed in the tropical Pacific east of the dateline (see Methods). With the same observed SSTAs, enhanced decadal SST variabilities off western Australian coast and in the Niño4w region are found in two sets of Pacific Ocean pacemaker experiments, albeit with slightly different timing for the emergence of linkage: since 1990 for CESM1.2 and since 1985 for FGOALS-f3-L (Figs. S2, S12). This confirms that the IPO has played a role in enhancing the decadal variabilities in these two regions.

Although the equatorial Pacific SST of each ensemble member is constrained by the same observations, the spread of the NNI between the ensemble members still exists (Fig. S13), suggesting that the IPO might not be the sole factor influencing the decadal variability in SST off western Australian coast. We found that the NNI amplitude shows a continuous increase since 1970 (Fig. 3a), especially for the decadal NNI amplitude (Fig. 3b). The enhancement of the decadal NNI is not consistent with the IPO phase transition in the observations (Fig. 3a, b). Meanwhile, the stable decadal linkage had already appeared (Fig. 1f)

before the IPO turned to its negative phase. Thus, the emergence of the interbasin (Indo-Pacific) decadal linkage since the 1980s cannot be attributed solely to the IPO.

The strengthened easterly wind anomalies across the western–central tropical Pacific over the past several decades are linked to the tropical Indian Ocean warming[10,25]. This rapid warming inspires us to explore the role of the tropical Indian Ocean in enhancing the decadal linkage between the western Australian coast and the western–central tropical Pacific (denoted as NNI-Niño4w linkage). Two sets of CESM1 pacemaker experiments are used to examine the effects of the tropical Indian Ocean SST on the NNI-Niño4w linkage: One is TIOGA (tropical Indian Ocean Global-Atmosphere) experiment in which the model SSTAs are restored to observed ones in the tropical Indian and western Pacific (west of the dateline), and the other, IOGA (Indian Ocean Global-Atmosphere) experiment in which the model SSTAs are restored to observations in the full Indian Ocean basin north of 50°S (see Methods). Results from TIOGA experiments reveal that the changes in the tropical Indian Ocean and part of western Pacific can generate significant SST changes in the central-western tropical Pacific and off western Australian coast (Fig. S14), accounting for a large fraction of the observed NNI after 1985 (Fig. 3c); Before 1985, however, the simulated NNI has a weaker amplitude and opposes to the observation (Fig. 3c). The simulated NNI and Niño4w index in TIOGA are out of phase with a correlation coefficient of −0.74 (Fig. 3d). These results suggest that SSTA in tropical Indian Ocean and western Pacific play an

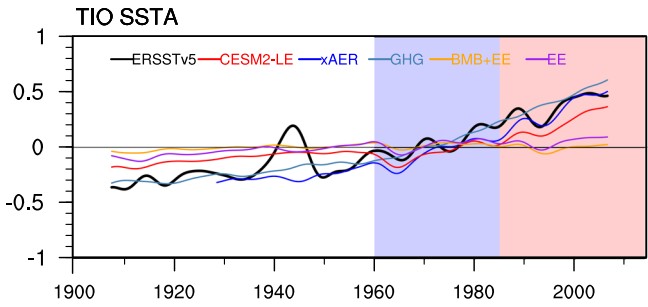

**Fig. 4 | The contribution of single external forcing to the Tropical Indian Ocean (TIO) sea surface temperature (SST).** The 8 years low-pass filtered tropical Indian Ocean Sea Surface Temperature Anomaly (SSTA; 15°N–15°S, 50°–160°E) in ERSSTv5 (black) and the ensemble mean of CESM2-LE (red) and four sets of CESM2 single forcing large ensembles: External forcings that exclude anthropogenic aerosols (xAER; blue), greenhouse gases only (GHG; light blue), biomass burning aerosol only and everything else (BMB and EE; orange), and EE (purple). Light blue shading identifies the period 1960–1985. Light red shading identifies the period after 1985.

important role in causing the enhanced NN-Niño4w linkage in recent decades. This is because a warm (cold) SSTA in the tropical Indian-western Pacific basin drives equatorial easterly (westerly) wind anomaly in the central-western tropical Pacific[25] (Fig. S15), enhances (reduces) equatorial upwelling and causes cold (warm) SSTA in Niño4w region. The enhanced equatorial easterly (westerly) wind anomaly increases (reduces) the Indonesian Throughflow, strengthening Ningaloo Niño (Niña) events. Meanwhile, the colder (warmer) SSTA in the Niño4w region can induce cyclonic (anticyclonic) circulation in the Southeast Indian Ocean, together with local air-sea feedback, further strengthening Ningaloo Niño (Niña) magnitudes[7,29,37,32]. Moreover, the Ningaloo Niño (Niña) SSTA can also provide positive feedback to the equatorial zonal wind anomalies and SSTA in Ninño4w region[7].

Note that NNI and Niño4w index are out of phase in TIOGA experiments both before and after 1985 (Fig. 3d), whereas the simulated NNI agrees with the observed one only after 1985 (Fig. 3c). This is because before 1985, the Niño4w SSTA generated by Indian Ocean SST is out of phase with that of the IPO, whereas after 1985 they are generally in phase (Fig. S16b). The warm (cold) Indian Ocean SSTA and negative (positive) IPO act in concert to intensify Niño4w SSTA and thus NNI after 1985, while the Indian Ocean SSTA weakens the IPO effect before 1985 (Fig. 3f), causing the enhanced decadal linkage between NNI and Niño4w index after 1985 as shown by observations.

To understand what causes the decadal variability of Indian Ocean SST, we analyzed the results from large ensemble experiments using CESM2 and MPI climate models (Fig. 3e). The ensemble means of these experiments show that external forcing is the major cause for the decadal SSTA of the tropical Indian Ocean after 1985 (Fig. 3e), whereas IPO is the major cause before 1985 with a positive (negative) IPO inducing warm (cold) SSTA over the Indian Ocean[25] (Fig. 3f). Upon examining the impacts of various single external forcings (Fig. 4), we show that the increase in greenhouse gases (GHGs; Fig. 4, light blue line) has significantly contributed to the recent trend of rapid warming in the Tropical Indian Ocean. External forcings that exclude anthropogenic aerosols (experiment xAER; Fig. 4, blue line) produce a stronger warming trend, which means anthropogenic aerosols somewhat decrease the warming trend in the Indian Ocean. In addition to a warming trend, experiment xAER also generates decadal variability in the tropical Indian Ocean SST (Fig. 4, blue line), mainly owing to the forcings by volcanoes[13] (included in everything else experiment, EE; Fig. 4, purple line). This finding suggests that external forcings by volcanoes is the main cause for decadal TIO SSTA and result in its out-of-phase relationship with IPO after 1985.

## Discussion

In this study, by analyzing multiple observational datasets, four sets of pacemaker experiments, two large ensemble simulations with 100 members, and four single forcing large ensembles, we found that the decadal variability in SST off western Australian coast and in the western–central tropical Pacific has enhanced since the 1980s. The magnitude of decadal variability is comparable with those on inter-annual timescales. The enhanced variabilities in these two regions covary on decadal timescale, building a close decadal linkage. After 1985, volcanic activity significantly influenced decadal SSTA in the Tropical Indian Ocean, leading to the enhanced decadal linkage between these two regions. The externally forced decadal SSTA in the tropical Indian Ocean can largely explain the observed SSTA after 1985 but not before 1985, because before 1985, tropical Indian Ocean SSTA follows the IPO. Due to nonlinearity in the coupled climate system, it is possible that the increased SST over the Indo-Pacific warm pool due to greenhouse gases and volcanic activities can generate decadal variability in convection and SST. Proof of this effect, however, is beyond the scope of this paper, but it is an important part of our future research.

The decadal SST in the tropical Indian and Pacific Oceans are out of phase (in phase) after (before) 1985 on decadal timescale, with a warm Indian Ocean associated with a cold (warm) Pacific after (before) 1985. Both the warm Indian Ocean and a cold Pacific cause strengthened easterly wind anomalies, resulting in a colder SST in the western–central tropical Pacific since 1985. The strengthened wind anomalies can push more warm water mass to the southeast Indian Ocean, leading to an increased heat advection. The cold SSTA in the western–central tropical Pacific induce the cyclonic wind anomaly in the southeast Indian Ocean, leading to the strengthened heat advection into the Ningaloo Niño region and the enhanced Leeuwin Current, which further increases heat advection, and resulting in a warm SSTA off western Australian coast. Meanwhile, the western Australian coast can also influence western–central tropical Pacific, ultimately inducing a close linkage between them. In heat budget analysis, the surface heat fluxes vary substantially with the datasets, especially during the boreal winter when the Ningaloo Niño develops[38]. However, the increased heat advection is supported by different datasets and methods. Additionally, although our study focuses on the enhanced linkage on decadal timescale, there is also an enhanced linkage on interannual timescale, between the western Australian coast and the western–central tropical Pacific. The possible interaction between linkages on interannual and decadal time-scales requires further research.

This study shows the enhanced decadal linkage and its impact on both regions. We demonstrate the role of the Tropical Indian Ocean SSTA in this linkage, but previous studies show that local processes off western Australian coast, such as cyclonic anomalies, are also important. On the decadal timescale, the negative IPO can induce positive sea surface height (SSH) anomalies in the equatorial western Pacific and the southeast Indian Ocean. Associated with this, there is a significant negative correlation between sea-level pressure (SLP) anomalies off western Australian coast and SSH anomalies in the equatorial western Pacific and southeastern Indian Ocean on decadal timescale[29]. Another view, the rapid warming in the Indian Ocean, which can also induce basin-wide negative SLP anomalies[39], is mainly controlled by increased anthropogenic forcings. This basin-scale SLP anomaly would induce cyclonic wind anomalies and impact the local processes off western Australian coast.

When the tropical Indian Ocean experiences significant warming, it induces negative SLP anomalies extending eastward to the western tropical Pacific. This process amplifies the zonal SLP gradient across the Pacific, which, in turn, generates anomalous easterlies in the tropical western Pacific[39]. Nevertheless, a common shortfall among climate models is their inability to accurately represent this dynamic

process. Specifically, these models often underrepresent the discrepancy in warming rates between the tropical Indian Ocean and the tropical Pacific. The tropical Indian Ocean warms at a faster rate than the tropical Pacific, a critical factor that most models fail to capture. This model bias results in an underestimation of the strong negative SLP anomalies. Consequently, this modeling bias could potentially influence the simulated SST off western Australian coast and in the western-central tropical Pacific.

Regarding the decadal linkage, the SST variability off western Australian coast and in the western–central tropical Pacific can be important indicators of the decadal variability in the Indo-Pacific coupled system. The warming in the tropical Indian ocean can also be affected by other interdecadal variabilities (such as the Atlantic Multidecadal Oscillation, AMO[9]). The warmer North Atlantic Ocean (positive AMO) can lead to the warming in the Indian Ocean[40] and Indo-Pacific warm pool[9]. The Indo-western Pacific SST response to the tropical Atlantic warming is almost immediate[41]. This indicates that there is a connection among three tropical ocean basins. Since the Indian Ocean warming (including Indo-Pacific warm pool) will continue[42], enhanced SST variability off western Australian coast could persist for decades into the future. The potential warming effect has a large chance to exceed the impact of the IPO on this decadal change in the Indo-Pacific coupled system, which has implications for modulating decadal predictions and managing the climate risks of marine heatwave events.

## Methods

### Observation datasets
The monthly SST data from 1900–2020 used in this study are obtained from the Extended Reconstructed Sea Surface Temperature dataset, version 5 (ERSSTv5)[43], the Hadley Centre Sea Ice and SST dataset (HadISST1)[44], and the Institute of Atmospheric Physics (IAP) analysis dataset (1940–2020)[45,46]. We use the monthly SST data (1900–2019) from Extended Reconstruction Sea Surface Temperature Version 3b (ERSSTv3b)[47], which excludes satellite data. The surface wind and 200-hPa wind data are from the National Centers for Environmental Prediction/National Center for Atmospheric Research (NCEP/NCAR) reanalysis dataset (1950–2020)[48]. The precipitation, latent heat flux, shortwave radiation, longwave radiation, and sensible heat flux data are from the Japanese 55-year Reanalysis (JRA55)[49] from 1958–2015. The Potential temperature, Zonal velocity, and Meridional velocity are from Ocean Reanalysis System 5 (ORAS5)[50] global ocean reanalysis monthly data from 1958–2015. All datasets are interpolated to $1° \times 1°$ grids.

### Simulation datasets
The effects of interbasin linkage are investigated using the results from Pacific pacemaker experiments with two coupled general circulation models: the Community Earth System Model, version 1 (CESM1)[51] from NCAR, and the low-resolution version of the Chinese Academy of Sciences (CAS) Flexible Global Ocean–Atmosphere–Land System model, finite-volume version 3 (FGOALS-f3-L)[52]. The model SSTs for the tropical central-eastern Pacific (20°N–20°S, 175°E–75°W) are restored to the model's climatological mean plus the observed anomaly (HadISST1). Tropical Indian Ocean Global-Atmosphere (TIOGA)[39,53] using CESM1, where SST in the tropical Indian Ocean and part of the western Pacific warm pool region is restored to observed SST from Extended Reconstructed SST version 3b (ERSSTv3b), with fully restored SSTA for 15°N–15°S, from the African Coast to 161°E, and buffer zones at 15°N–20°N, 15°S–20°S, and 161°E–180°. Another experiment is IOGA (full Indian Ocean Global-Atmosphere)[54] using CESM1, where SST in the Indian Ocean (north of 50°S) is restored to observed SSTA from ERSSTv3b. Meanwhile, the oceanic and atmospheric components are freely coupled in other ocean areas[55]. Historical radiative forcing is used to drive CESM1 pacemaker

(CESM1 POGA, with eight members) for 1870–2005 (is extended to 2014), and FGOALS-f3-L pacemaker (FGOALS-f3 POGA, with ten members) for 1870–2014. TIOGA (ten members) is from 1920 to 2013. IOGA (ten members) is from 1920–2019.

In addition to the pacemaker experiments, two large ensembles for the 1950–2014 period are also used: the 100-member Community Earth System Model version 2 Large Ensemble (CESM2-LE)[56], and the 100-member Max Planck Institute for Meteorology Grand Ensemble (MPI-GE)[57] to explore the effects of the external forcings on the enhanced decadal variability. To further isolate the effects of different external forcings on these decadal variability, four CESM2 single forcing large ensembles[58] are analyzed: greenhouse gases only (GHG, 15 members); biomass burning aerosol only (BMB, 15 members); everything else (EE, excluding greenhouse gases, and anthropogenic aerosols; 15 members); and excluding anthropogenic aerosols (xAER, i.e., all forcings are time-evolving except anthropogenic aerosols, 10 members). In those "only" experiments, the time-evolving forcing is only the named one, and all other external forcings are fixed at the 1850 level. In EE and xARE experiments, all forcings are time-evolving except those excluded forcings. other than those that are time-evolving in GHG or BMB are time-evolving. The data we used is from 1900 to 2014 except xAER which the data is only available between 1920 to 2014.

### Definition of indices
The SST field is regressed onto the global mean SSTA for detrending to remove the influence of anthropogenically induced global warming[59], and the residual terms of the regression are used for analysis. The area-weight-averaged SSTA in the wedge-like area[7,32] (22°S–16°S, 102°E–108°E; and 32°S–16°S, 108°E–115°E; see blue box in Fig. 1a) is used to define the NNI. This definition can comprehensively capture the warming pattern off western Australian coast as being wedge-shaped. Compared with the traditional definition[5,21] whose area-weight-averaged SSTA is based on the region 110°–116°E, 22°–32°S, similar NNI variation and amplitude (denoted as STD) are obtained.

Ensemble empirical mode decomposition (EEMD)[60] is applied to isolate components of the NNI and the Niño4w index (5°N–5°S, 160°E–175°W; see red box in Fig. 1a). The added white noise in each of the 1000 EEMD ensemble members had a standard deviation of 0.2[61]. The NNI is decomposed into seven different timescale components. The power spectra of each component are calculated to obtain the timescale. According to the power spectrum, the third and fourth components are combined as interannual signals for 1–7 years. The fifth and sixth components are combined as decadal signals with periods of more than seven years. The IPO index is following the Tripole Index definition[36], the SSTA used is from region 1: 25°N–45°N, 140°E–145°W; region 2: 10°S–10°N, 170°E–90°W; and region 3: 50°S–15°S, 150°E–160°W. By using the 8–16-year band-pass filter, we extracted decadal fluctuations in indices, such as the decadal NNI. In analyzing the mixed-layer heat budget, we calculated these decadal heat budget terms and also determined average values based on the development period of the decadal Ningaloo Niño events, before and after 1985. We select periods from when the decadal NNI increased from 0 to its peak, corresponding to the development period of the decadal Ningaloo Niño events. Before 1985, the development periods are from December 1970 to December 1972 and from January 1981 to August 1983; while after 1985, the periods are from June 1986 to June 1988 and from October 1995 to August 1997.

### Analysis for the formation of SST patterns
To explore the mechanisms underlying the formation of the SST patterns, we employ wind–evaporation–sea surface temperature (WES) feedback analysis[33,34]. According to ref. 34, changes in ocean heat transport ($D'_0$) and net surface heat flux are balanced against each

other on long timescales:

$$C \frac{\partial T'}{\partial t} = D'_0 + Q'_{net} \tag{1}$$

where the left-hand side of Eq. (1) equals zero; $T'$ is the SST change; and $Q'_{net}$ is the net surface heat flux anomaly ($Q'_{net} = Q'_{sw} + Q'_{lw} + Q'_{sh} + Q'_{lh}$; positive downward), which consists of shortwave radiation ($Q'_{sw}$), longwave radiation ($Q'_{lw}$), sensible heat flux ($Q'_{sh}$), and latent heat flux ($Q'_{lh}$). Furthermore, the latent flux is decomposed into a Newtonian cooling $Q^{o'}_{lh}$ component and a residual that represents atmospheric forcing $Q^{a'}_{lh}$:

$$Q^{o'}_{lh} = \alpha \overline{Q_{lh}} T' \tag{2}$$

$$Q^{a'}_{lh} = Q'_{lh} - Q^{o'}_{lh} \tag{3}$$

The equation of SST pattern can be written as follows:

$$T' = \frac{D'_0 + Q'_{sw} + Q'_{lw} + Q'_{sh} + Q^{a'}_{lh}}{\alpha \overline{Q_{lh}}} \tag{4}$$

$\alpha$ is -0.06 K$^{-1}$ in the tropics. The WES feedback is represented by $Q^{a'}_{lh}$ and $D'_0$, showing three-dimensional oceanic temperature advection.

### Analysis of the mixed-layer heat budget
We calculate the mixed-layer heat budget, and the equation is as follows[62,63]:

$$\frac{\partial T'}{\partial t} = - \left( \bar{u} \frac{\partial T'}{\partial x} + u' \frac{\partial \bar{T}}{\partial x} + u' \frac{\partial T'}{\partial x} \right) - \left( \bar{v} \frac{\partial T'}{\partial y} + v' \frac{\partial \bar{T}}{\partial y} + v' \frac{\partial T'}{\partial y} \right) + \frac{Qnet'}{\rho_0 C_P h} + R' \tag{5}$$

T, u, and v represent the oceanic temperature, zonal and meridional current velocities averaged in the mixed-layer depth, respectively. $\frac{\partial T'}{\partial t}$ is temperature tendency. A variable (such as $T$, $u$, and $v$) can be decomposed into the sum of the climatological mean state (e.g., $\bar{u}$) and anomaly state (e.g., $u'$)[64]. $Qnet'$ is the "net surface heat flux", which accounts for the sum of shortwave radiation, longwave radiation, latent heat fluxes, and sensible heat fluxes. $\rho_0$ indicates sea water density (1026 kg/m$^3$) and $C_P$ is the ocean heat capacity (3986 J/(kg · K), $h$ is mixed-layer depth. $R'$ is the "residual" term, which accounts for the remaining heat budget components not explicitly resolved by the used data in the equation. The results presented above are obtained through the vertical integration of the heat budget across the mixed layer.

To calculate the mixed-layer depth, we follow[65]:

$$\delta_k = \sqrt{\frac{1}{k} \sum_{i=1}^{k} (\lambda_i - \overline{\lambda_k})} \tag{6}$$

$\delta_k$ is the standard deviation, $\lambda(z)$ represents the temperature, and z is the depth of data. When k equals 1, it is on the sea surface. $\overline{\lambda}_k$ is the mean between $z_1$ and $z_k$.

The maximum variation over the same depth range is:

$$\sigma_k = \max_{i=1}^{k}(\lambda_i) - \min_{i=1}^{k}(\lambda_i) \tag{7}$$

The relative variance of $\lambda$ between $z_1$ and $z_k$ is:

$$\chi_k = \frac{\delta_k}{\sigma_k} \tag{8}$$

When $\chi_k$ gets smaller, the $z_k$ reaches $z_{n1}$. $z_{n1}$ is the layer below and slightly deeper than the mixed layer. Therefore, we locate $z_{n2}$ by the following:

$$\frac{\delta_{n2+1} - \delta_{n2}}{\delta_{n1+1} - \delta_{n1}} \leq \Delta \tag{9}$$

Usually, the value of $\Delta$ is between 0.1 and 0.5, and we use 0.5. $z_{n2}$ is the mixed-layer depth.

### Statistical analysis
The wavelet transform is calculated based on the Morlet function of the NNI and the Niño4w index to test whether they show similar periodicities. The sliding correlation and coherence spectrum between the NNI and the Niño4w index are also calculated. We use regression and correlation maps between seasonal mean variables (e.g., the SSTA and precipitation anomaly averaged for December-February) and the decadal NNI based on DJF mean to visualize the spatial pattern of the influence of the tropical Pacific on the region off western Australia, as previous studies have indicated that the Ningaloo Niño reaches its peak during boreal winter[5,21,29,32]. Additionally, singular value decomposition (SVD) is employed to demonstrate the relationship between SST and convective activity.

The wavelet results are tested statistically against red noise[66]. The statistical significance of the correlation coefficients is assessed using the two-tailed Student's t-test. Given our application of low-pass filtering and similar smoothing methods over decadal periods, which reduces degrees of freedom, we utilize effective degrees of freedom in the test[67]:

$$\frac{1}{N_{eff}} \approx \frac{1}{N} + \frac{2}{N} \sum_{j=1}^{N-2} \frac{N-j}{N} \rho_{XX}(j) \rho_{YY}(j) \tag{10}$$

where $N$ is the sample size, $\rho_{XX}(j)$ and $\rho_{YY}(j)$ are the autocorrelation functions of two time series $X$ and $Y$ at time lag.

### Reporting summary
Further information on research design is available in the Nature Portfolio Reporting Summary linked to this article.

## Data availability
All observational and model datasets used in this study are available publicly or on request. Observational data are available from NOAA (https://psl.noaa.gov/data/gridded/data.noaa.ersst.v5.html and https://climatedataguide.ucar.edu/climate-data/sst-data-noaa-extended-reconstruction-ssts-version-3-ersstv3-3b), the Hadley Centre (https://www.metoffice.gov.uk/hadobs/hadisst/), the NCEP/NCAR Reanalysis Project (https://psl.noaa.gov/data/reanalysis/reanalysis.shtml), and the Japan Meteorological Agency (https://jra.kishou.go.jp/JRA-55/index_en.html). The Data from IAP observation is available via http://www.ocean.iap.ac.cn/pages/dataService/dataService.html. The data from ORAs5 is available via https://cds.climate.copernicus.eu/cdsapp#!/dataset/reanalysis-oras5?tab=form. The CESM2 large ensemble datasets are available via https://www.cesm.ucar.edu/projects/community-projects/LENS2/data-sets.html. The MPI large ensemble datasets are available via https://esgf-data.dkrz.de/projects/mpi-ge/. The four single forcing large ensembles are available via https://www.cesm.ucar.edu/working-groups/climate/simulations/cesm2-single-forcing-le. The tropical Indian Ocean pacemaker experiment (TIOGA) datasets are available via https://www.cesm.ucar.edu/working-groups/climate/simulations/cesm1-indian-ocean-pacemaker. The related data of some pacemaker experiments (CESM1 Pacific pacemaker experiment, FGOALS-f3 Pacific pacemaker experiment, and Indian Ocean pacemaker experiment) are available via https://doi.org/10.57760/sciencedb.07476. Source data are provided with this paper.

## Code availability

The codes for the figures can be downloaded at https://doi.org/10.57760/sciencedb.07476.

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

## Acknowledgements

This study was supported by the Key Program for Developing Basic Sciences (Grant No. 2020YFA0608902 for PL and YD and 2022YFC3104802 for HL), National Natural Science Foundations of China (Grant Nos. 41976026 for PL, 41931183 and 92358302 for HL, and 42376021 for LZ), the Strategic Priority Research Program of Chinese Academy of Sciences (XDB42000000 for YL; XDB42010404 and XDB0500303 for PL and HL), and the development fund of South China Sea Institute of Oceanology of the Chinese Academy of Sciences SCSIO202203 for LZ. This work is partially supported by the Regional and Global Model Analysis (RGMA) component of the Earth and Environmental System Modeling Program of the US Department of Energy's Office of Biological & Environmental Research (BER) via the National Science Foundation (NSF) IA 1947282 (DE-SC0022070) for AH. The National Center for Atmospheric Research is sponsored by the NSF of the United States of America under Cooperative Agreement No. 1852977 for AH. The work was also supported by the Natural Science Foundation of Jiangsu Province (BK20230061) for LC. The work also acknowledged the technical support from the National Key Scientific and Technological Infrastructure project "Earth System Science Numerical Simulator Facility" (EarthLab) and Oceanographic Data Center, Institute of Oceanology, Chinese Academy of Sciences.

## Author contributions

Y.D. shaped the idea, performed the analysis, and drafted the manuscript. P.L. designed the research, provided comments, and revised the manuscript. W.H. helped developing the idea. H.L., B.W., Y.L., L.C., and A.H. helped organize and revise the draft. Y.L. and L.Z. provided comments and contributed to the discussion of the results. L.Z., W.H., Y.W., B.Z., W.B., and Y.Y. performed the numerical simulations; All authors contributed to interpreting the results and improving the text.

## Competing interests

The authors declare no competing interests.
