## [Peer Review File · Nature Communications]

Emergence of Decadal Linkage between Western Australian Coast and Western-Central Tropical PacificREVIEWER COMMENTS

Reviewer #1 (Remarks to the Author):

Recommendation: Minor revisions

The paper discusses the inter-basin linkage between western Australian coast and western-central tropical Pacific on the decadal time scale (8-16 years) based on the analysis of multiple datasets, including three sets of observational data, two pacemaker experiments of climate models, a large ensemble of CESM2 simulations, and CMIP6 simulations. The results suggest that a decadal linkage of SST between the two regions emerged around 1985, coinciding with a continuous intensification of decadal variability. The mechanism of the decadal linkage is further discussed by examining the role of external forcing and the IPO. Warming in the western tropical Pacific induced by external forcing causes enhancement and eastward-shift of the center of deep convection from the Indian Ocean to Maritime Continent, leading to stronger westerlies in the tropical Indian Ocean. These changes cause anomalous cyclonic circulations and SST changes in the west coast of Australia, resulting in enhanced coupling in the two regions. These findings provide valuable insights into decadal variability over the Ind-Pacific region and its prediction.

While the availability and quality of observational data are generally challenges for this type of studies on decadal variability, the authors conclusions are obtained from the analysis that incorporates multiple datasets including both observations and model experiments. Therefore, the decadal linkage scenario described above are plausible, making it worthwhile to publish the results. I have some minor comments, which are mostly concerns on writing.

1. Title

“Western-Central Pacific” is not very clear. Perhaps “Western-Central Tropical Pacific” is better.

2. Abstract

L25 “The interbasin linkage generates a broad and profound impact.”

Adding a little more background information at the beginning of the abstract (perhaps two sentences) would help readers better understand the implication of the results.

L29 “external forcing”

In the discussion (L241), the term “the external forcing” is used to signify “anthropogenic forcing”. Using a more explicit expression or term related to global warming (e.g., increase of greenhouse gas) would clarify the meaning of “external forcing”.

3. L42 “the western Pacific (including eastern Indian Ocean...”

This sentence is confusing. The western Pacific does not include the eastern Indian Ocean.

4. L51 “*varia*”

Should be “*variation*”.

5. L54 “*most important*”

Important for what? Does this mean “*strongest*”?

6. L84 “*NNI, which is defined as the area-weighted mean SST anomaly (SSTA) in the wedged area*”

This definition is different from the most commonly used NNI (area average SST for 110°-116°E, 22°-32°S or 108°E-116°E, 22°-28°S). Are the results sensitive to the definition used? For example, is STD on the decadal time scale similar when using the commonly used NNI? The justification for the choice of this specific definition could be included in the method section.

6. L86. “*The standard deviations (STDs) of NNI are 0.92 °C (1950-1985) and 1.07 °C*

87 (1985-2020).”

The quality of the SST data could be different in these two periods because of the satellite measurements in recent years. Is the uncertainty due to the SST data quality smaller than the difference in STD between the two periods?

7. L93 “*8-16 years*”

It is beneficial to state more explicitly in this section that the decadal time scale in this study is “*8-16 years*”, although it is stated in the method section later.

8. L134 “*The center of negative anomaly shifts westward*”

Before 1985, strong correlations exist in the eastern tropical Pacific. However, this may not necessarily indicate an interbasin linkage between western Australian coast and eastern tropical Pacific during this period. Or is it an interbasin linkage? Please explain.

9. L145 “*eastern*”

Should be “*easterly*”.

10. L148 “*norther*”

Should be “*northerly*”.

11. L151 “*Weakening of coastal upwelling*”

Also, northerly winds reduce the wind speed and thus evaporative cooling (latent heat flux) decreases.

12. L159 “*The latent heat flux is largely unchanged between these two periods*”

Fig. 2 shows that local winds (and thus wind speed) are largely changed between these two periods. So the latent heat flux should be largely changed as well, which is inconsistent with the results. Please clarify.

12. L166 “*northern wind anomaly*”

Should be “northerly wind anomaly”.

13. L179 “centra-eastern”

Should be “central-eastern”

14. L205 “centra-eastern”

Should be “central-eastern”

15. L223 “This indicates external that”

Correct to “This indicates that external”

16. L245 “convection in situ”

Not clear what this means.

17. L255 “northern surface wind anomaly”

Should be “northerly surface wind anomaly”

18. L279 “eastern wind anomaly”

Should be “easterly wind anomaly”

19. L332 “Japanese 55-year Reanalysis (JRA55)”

The time period covered in OAFlux is similar. Is the decadal variability in surface flux data from JRA55 consistent with OAFlux? Also, the uncertainty of surface flux data in this region should be mentioned (e.g., Feng and Shinoda 2019).

20. Reference section

Most of the references lack journal names.

21. L592 Data and materials availability

Availability of two pacemaker simulations is not provided.

Reviewer #2 (Remarks to the Author):

Review of the paper entitled ‘Emergence of Decadal Linkage between Western Australian Coast and Western-Central Pacific’ by Dr Lin and colleagues.

Main

This work investigates the recent enhancement of decadal variability observed in the northwest Australian coast. Through observational data analyses and model experiments, the paper suggests that the anomalous warming of the western Pacific tropical Ocean contributes to enhanced variability via

atmospheric teleconnections. The Paper also suggests an emergence of decadal linkage between the two regions under the western Pacific warming which is likely to persist in the future given the projected warming of the tropical Pacific.

The role of western Pacific variability on the development of Ningaloo Nino (NN) has been reported earlier. Feng et al. (2015) also suggested that anthropogenic warming in the western Pacific may have an additional role in modulating the NN at decadal time scales. This paper takes this idea further with detailed research to show that recent decadal variations of NN might have an independent existence from IPO under strong external warming, leading to a 'new decadal link' between the two regions. This is an exciting finding!

I appreciate the enthusiasm of the authors to tackle a complex research problem that has narrow lines separating the role of anthropogenic warming and natural variations for controlling variability over different time scales. Given the complexities of the problem, I still have quite a few concerns and questions which I list below. I hope clearing these questions will bring some more clarity to the presentation of the paper.

Major comments:

1) What is the rationale behind using EEMD to obtain components of variability over different time scales? For instance, how do the authors make sure that the decadal component (fifth and sixth components combined) of indices has variability only above 7 years? Also, it seems that only indices are filtered for different time scales but not the variables (SST, wind, etc.). You may get spurious regression estimates if the dependent (SST for example) and independent (indices) operate over different time scales. I would like the authors to make a little more serious effort to clarify these aspects in the paper.

2) (Page 17) Analysis of SST pattern: This section doesn't speak of ocean heat transport (D) well. The balance between D and Q is set as an assumption but hasn't been tested explicitly. In fact, previous works on low-frequency NN and its mechanisms (Feng et al. 2015) showed that heat advection from the western Pacific plays an important role. The present paper summarises the WES and ocean transport contributions in the form of regression coefficients (table S2) which is very hard to understand. I do not even understand what are the two sets of values presented for each period in this table. I find that the way of treatment of ocean heat transport is an important drawback of this study. In fact, there is no description of how and where heat transport anomalies are calculated. Atmospheric teleconnections might be playing an important role as discussed in this paper, but I would like to see if the role of oceanic heat transport can be expelled more undoubtedly. For instance, there might be a reason why ITF doesn't contribute much to the decadal linkage, however, this study doesn't explore that.

3) Related to the previous comment: The role of (slow) upper ocean heat content variations in the western Pacific and its transmission to the West Australian coast (WAC), affecting SST and sea level there, is indeed discussed earlier (Feng et al. 2015). This paper almost fully neglects the role of ocean waveguides in shaping the low-frequency SST variations in the WAC. I was wondering how tide-gauge sea-level measurements at Fremantle which is identified to be a good indicator of heat content variations in the WAC (Feng et al. 2003, 2015) would complement the overall result of this paper. Even though the authors try to establish the absence of oceanic teleconnection (S3?), the description of figure 2 (Page 7) uses the ocean transport (Leeuwin current) to explain the spatial structures. Do authors

believe that the influence of Leeuwin current is confined only to the NN region?

4) There is a systematic difference between the NNI-regressed spatial SST maps in Figure 1 and Figure 3. There is no variability seen in the western Pacific in figure 1 (observation) but there are strong positive SST anomalies in the western Pacific in figure 3 (coupled model). Can the authors elaborate on this difference further? An intercomparison of figures 1, 3, and S8 prompts me to think that the NN in coupled models is tightly linked to canonical ENSO and affected by its interannual variations but this is not the case for observation. This may have strong implications for the overall result presented in this paper. For instance, in the pacemaker experiments, the ENSO variations dominate and the effect of Maritime continent warming after 1990 might be nil or weak. But, in observations, Maritime warming might play an important role in shaping the decadal linkage. I would ask the authors to redo figure S5 but for models (pacemaker experiments) for the two periods to see this further.

5) Related to the previous comment - (Page 10) authors say 'surprisingly, the nni-nino4w decadal linkage is found in the ensemble mean of members whose IPO phases are totally opposite to observed phase' – this may simply indicate that, in the ensemble mean, it is not IPO but interannual variability in the equatorial Pacific affects the linkage. In fact, Fig. S11 suggests that there is virtually no decadal power in the ensemble mean Nino4w for R-, indicating that the spatial pattern seen in figure 3f comes from interannual variability. This is contradictory to the argument made, and I would ask the authors to make a revision accordingly either for the analysis or the text at the beginning of Page 11. Also, I would like to ask authors to show spatial maps similar to Figures 3e, f but for the entire 100-member ensemble mean SST (not for R+/- groups).

6) Figure 3: Do the wavelets represent the ensemble mean spectrum or the spectrum of the ensemble mean index? I think these two are different. Also, there is a 4-year periodicity in the Nino4w in figure 3d which is absent in figure 1e. Shouldn't the ensemble mean spectrum eliminate such interannual signals? Can authors explain why this is so?

Other comments:

1) Page 4; line 71: Do we have enough observations before 1980 in the western Australian coast to confidently state a post-1980 change? I would like to hear from the authors if they think the enhancement of decadal variations since 1980 is because we have robust measurements since 1980 (satellite) in this region.

2) Statistical significance of the coherence over a 21-year period and other regression estimates. The methodology says the test employs a student's t-test but doesn't explicitly state how the effective sample size (ESS) is estimated. Since the signals are smoothed over decadal periods, the autocorrelation of the indices/time series would reduce the independent samples, and significance tests should consider the ESS.

3) The IPO definition doesn't follow the reference (Henley et al. 2015) given. Henley et al. suggested a tripolar index instead of EOF.

4) Introduction:

Page 3; line 45: 'Variations in the interbasin linkage between off western Australian coast and the western–central Pacific, linked by the Indo-Pacific warm pool and the overlying atmospheric circulation, can generate a profound impact on the local and remote climate'.
'Better rephrase to 'can impact local and remote climate'.

5) Page 3; line 50: Not discussing 'local' processes well: How much variance is explained by local and remote processes respectively? Similarly, what is the main mode of low-frequency (interannual and above) SST variability on the West Australian coast? Is it Ningaloo Nino or warming associated with the Leeuwin current?

6) Page 8; line 172: The convective centers change over P1 (IO) and P2 (WEP). Does that really mean an 'eastward movement' as suggested in the paper?

7) Page 11, and elsewhere in the paper: The physical explanations presented in the paper suggest that the western Pacific warming (anthropogenic) can enhance an 'existing' decadal variability in the WAC – but doesn't induce a new mode of decadal variability. In that perspective, is it feasible to say an 'emergence of decadal linkage'? More appropriate would be to say, 'influence of western Pacific warming on the amplitude of decadal variability off the WAC'?

8) Fig. 4: Why does the trend estimation is limited to 1970 – 1995 for CMIP6 models? Is the phase of the IPO in CMIP models consistent with the observed IPO phase? If not necessarily (I think that is the case), how would the authors reject the fact that the linear relationship between the western Pacific SST and NNI amplitude might arise through IPO? I would suggest the authors add one more panel by redoing the same figure but once the IPO is regressed out from the western Pacific SST, to check this.

9) Pacemaker experiment – The SST anomalies are nudged to the model climatology and HadISST1 anomalies. I would like to see the climatological SST map in the Indo-Pacific region from HadISST1 and the two models considered for the pacemaker experiment (left) and the spatial map of the standard deviation of the SST anomalies (for HadISST and two models) for the two periods (post and before 1980) in the right columns. And, please show the Nino4w & NN regions (box) over each of these maps.

Minor comments:

1) Previous studies indicate that half of the NN is NOT related to the western Pacific (La Nina), indicating an important local driving mechanism that controls wind and SST off the WAC (see Kataoka et al. 2013; Marshall et al. 2014). The respective role of local processes driving the enhanced decadal amplitudes is not well addressed in the present study. Can the authors include a few more notes about the anthropogenic effects on the local drivers, in the discussion?

2) Anthropogenic heat enters the ocean through natural variability (e.g., IPO). The global warming hiatus during the last decade is an example. It is hard to distinguish, hence, the decadal SST response to natural variability and external forcing over short periods. A coupled model experiment by Feng et al. (2015) indeed suggested that, even though external warming influences the preconditioning of the NN

development, it (external forcing) itself may not cause the decadal modulation of the occurrence of NN over recent decades – I would ask authors to discuss these aspects and previous literature in the discussion of the paper.

3) Please provide the time series of decadal (>7 years) Nino4w and NNI together over 1950 – 2020 and provide the correlation coefficient for the periods 1950 – 2020, 1950 – 1985, and 1985 – 2020.

Reviewer #3 (Remarks to the Author):

Based on observations together with several sets of model simulations, i.e., pacemaker simulations from CESM1.2 and FGOALS-f3-L models, single-model CESM2-LE large ensemble simulations, and CMIP6 multi-model simulations, the authors investigated the coherence in decadal variations of sea surface temperature (SST) between the western Australian coast and the western-central Pacific. The coherence of SST variations on the decadal time scale between these two regions emerged in 1985. The authors claimed that the enhanced covarying decadal variations were attributed to the rapid warming in the western Pacific, however, the influence of Interdecadal Pacific Oscillation phase changes is relatively less prominent. The topic is important and interesting. The manuscript is in general well-organized and written, but some parts especially the methods could benefit from greater clarity. The conclusions that attribute the coherent decadal variations in the western Australian coast and the western-central Pacific to global warming merit further analyses/discussions to bring this forward more robustly.

Major comments:

- The main conclusion and novelty of this study is attributing the decadal linkage between the western Australian coast and the western-central Pacific to the global warming induced by external forcing. This part of the results and discussion needs to be further enhanced. In the present manuscript, the authors use CESM2-LE large ensemble to investigate this and the ensemble mean of 100 members represents the forced signal. By comparing Fig. 3(c) and (d) based on CESM2-LE large ensemble mean to Fig. 1(d) and (e) based on observations, the decadal variations from the time scale of 8-16 years are quite different in the wavelet analysis results, especially the Niño4w index is mainly dominated by 3-7 year variations. The wavelet results in Fig. 3(c) and (d) don't have information of significance test as in Fig. 1(d) and (e). This needs to be added and it will be helpful to add the time series of the ensemble mean NNI and Niño4w from CESM2-LE.

- Fig. 4: What would the figure be like if the Y-axis was substituted with the correlation between NNI and Niño4w in the period before and after 1985? This would be more straightforward in supporting the enhanced decadal coherence between NNI and Niño4w due to external forcing-induced warming.

- The authors selected subsamples from CESM2-LE to investigate the influence of IPO on the enhanced covarying SST decadal variabilities in the western Australian coast and the western-central Pacific. It is

unclear how the IPO phases of the selected members match observations. The Methods section explained shortly that the members are selected based on the IPO pattern correlation coefficient between the model and observation. How are the time-evolving IPO phases combined with the spatial patterns? I would suggest simply selecting the ensemble members that could resemble the IPO index phase changes from observations.

- L295-298 “Most climate models present relatively even warming in the tropical PacificThis model bias may affect the emergence timing of Indo-Pacific decadal linkage,” What is the bias for CESM2? As CESM2-LE is crucial for this study, the warming pattern from this model should be presented.

- It seems that the main analyses are on boreal winter time, i.e., December-January-February (DJF). In some places, it is stated, and in others not. It would be much clearer if the authors could state in general if seasonal or annual data was used in this study at the beginning of the manuscript and in the Method part. In case all the analyses are using DJF data, this information could be considered to be added to the abstract and even the title of the study.

- The Methods are not explained very clearly.

i. The section “Analysis for the formation of SST patterns” is missing some information. The way leading from equation (1) to (2) is not well explained, and the Newtonian cooling component $Q_{lh}(\sigma^{\prime})$ is not found in equation (2).

ii. The amplitude of NNI on the decadal timescale is larger than that on the interannual timescale (refer to L95-96). Normally the interannual variability is more prominent with a larger amplitude than the decadal variability because the time series on the decadal time scale is more smoothed. I wonder how the amplitude is defined.

iii. The IPO pattern correlation coefficient (PCC) is not introduced clearly how it is calculated. This is critical for understanding the results.

- Please review and ensure the grammar is correct and keep the consistency in the use of either past or present tense throughout the manuscript.

Minor comments:

- The first sentence in the abstract is incomplete or lacks information. Some words could be added to the end of the sentence to specify what impact the interbasin linkage would bring.

- L42-43: “...has experienced the most rapid warming...” I feel some conditions should be added, for instance “in the tropical regions”, because the high latitude warming signal is stronger than the warming in these regions.

- L55: “...affect that in the Niño4w region...” What does the “that” refer to?

- L97: “in other areas around the region off western Australian coast,” this is duplicated shown another time by the end of the sentence.

- L135-136: “Note that the positive anomaly appears stronger in the southern South China Sea and the western Pacific”, what does that mean?

-L148: “the norther wind...”-> “the northern wind...”

-L166: “the dynamic process has led to...” -> “the dynamic process via ocean heat transport has led to...”

-L173: “winter NNI and precipitation” -> It sounds a bit weird because the location of the NN region is in the southern hemisphere, the winter there is not in DJF. To keep consistent and be more precise, either stating the months or referring to the hemisphere would work.

-L182-183: “...more in accord with their position in the climatological mean state...” Adding a figure of the climatological mean state will make it clearer.

-L204: “Two sets of pacemaker experiments” Is it intended to say “with two models”?

-L209: Why is the time period for CESM1.2 shorter than that for FGOALS-f3-L?

-L222: “is similar to that observed exists for NNI...” -> “is similar to that observed for NNI...”

Actually, the decadal variability seems not similar between model simulations and observations, see my major comment.

-L225-226: “The external forcing explains about 42.6% of the decadal variance of the NNI.” Where does this number come from? It is better to show the time series.

-L233-234: “The decadal linkage is consistent in the individual members due to large Signal-to-Noise.” I don’t understand this sentence.

-L245: “convection in situ”-> “in situ convection”

-L250: “correlation coefficient (0.6) between the decadal NNI and the warming trend”, what would the value of correlation coefficient be when the decadal NNI is substituted with the correlation between decadal NNI and Nino4w?

-L303: “by Feng et al.24 and Trenery and Han26.” Why not just add the numbers as citations?

-L308: “It is unnecessary to cite the paper here again. It reads like reference #37 is a previous publication of the authors. BTW, the information on this paper in the reference list is incomplete.”

-L345: “with eight simulations” do you mean “ensemble members”?

-L346: “ 31 simulations” models? Simulations from 31 Earth system models?

L359: “ 1000 EEMD ensemble members ...0.2. ...” Is 1000 a typical number of members? What does the magnitude of standard deviation of 0.2 mean to the results? How was the number determined, would a smaller value of 0.1 or a bigger value of 0.3 work?

L366: “by dividing each monthly anomaly by the standard deviation”-> “by dividing each monthly anomaly with the standard deviation”

L369: to have the full name of WES

L370: the citation should be #29, the #32 is not Xie et al.

L377: $Q_{lh}(\sigma^1)$ This term does not appear in the equation (2)

L381: “where Q_{lw}^1 and Q_{sh}^1 are neglected to owe to their..” This is confusing. These two terms are presented in Table S2. Is this statement based on the results there? Then it should be in the main text. This sentence is not grammarily correct.

L390: decadal NNI Is NNI also based on DJF mean or annual mean data?

L392-393: From observation, we choose the CESM2-LE members with IPO phases similar (R+) /opposite (R-). I don’t understand this. Maybe the authors meant “According to” or “Based on” observation? “with IPO phases similar (R+) /opposite (R-).” -> “with similar (R+) /opposite (R-) IPO phases.”

L393-394: “Before choosing” could be removed.

L394-395: “PCC” is only used here and in the next sentence, I would suggest not introducing a new abbreviation.

Fig. 1: There is a typo in the title of Fig. 1(g): “Corss” -> “Cross”.

Fig. 2: Regarding the decadal NNI, there are only 3-4 cycles within 35 years, what is the degree of freedom for the statistically significant test?

The precipitation contours in Fig. 2(c)-(d) are hard to find and the values are missing.

“Ningaloo Niño generally peaks in boreal winter; thus, only the changes in boreal winter are presented.”

-Is this a general statement that could be applied to the whole study and all the figures shown in the manuscript? In that case, this needs to be mentioned at the beginning of the manuscript and introduced in the methods.

Fig. 3: The way of choosing the R+/R- group in this study might be affected by the model’s response of SSTA to IPO. It's possible that some SSTA patterns may not be related to certain IPO phases. As the purpose is to check the IPO, it is more straightforward to choose the ensemble members by the simulated IPO indices. Would this way end up with two similar sets of ensemble members?

Fig. 4: I would suggest adding figures with substituting the y-axis with correlation coefficients between NNI and Nino4w before and after 1985.

Table S2: It will be easier to add the long names of the terms into the table after each term.

Reviewer #1 (Remarks to the Author):

Recommendation: Minor revisions

The paper discusses the inter-basin linkage between western Australian coast and western–central tropical Pacific on the decadal time scale (8–16 years) based on the analysis of multiple datasets, including three sets of observational data, two pacemaker experiments of climate models, a large ensemble of CESM2 simulations, and CMIP6 simulations. The results suggest that a decadal linkage of SST between the two regions emerged around 1985, coinciding with a continuous intensification of decadal variability. The mechanism of the decadal linkage is further discussed by examining the role of external forcing and the IPO. Warming in the western tropical Pacific induced by external forcing causes enhancement and eastward-shift of the center of deep convection from the Indian Ocean to Maritime Continent, leading to stronger westerlies in the tropical Indian Ocean. These changes cause anomalous cyclonic circulations and SST changes in the west coast of Australia, resulting in enhanced coupling in the two regions. These findings provide valuable insights into decadal variability over the Ind-Pacific region and its prediction.

While the availability and quality of observational data are generally challenges for this type of studies on decadal variability, the authors conclusions are obtained from the analysis that incorporates multiple datasets including both observations and model experiments. Therefore, the decadal linkage scenario described above are plausible, making it worthwhile to publish the results. I have some minor comments, which are mostly concerns on writing.

Response: We would like to thank the reviewer for the constructive review comments/suggestions, which give us a big help to improve the quality of our manuscript. We have done our best to address the reviewer's concerns and modified the manuscript. Point-by-point responses to the reviewer's comments are listed below. We made our effort to correct errors in the writing, discussed the impact of uncertainties in heat flux data on the SST pattern off the western Australian coast. We also show that the uncertainty of SST data due to satellite measurements does not affect the decadal variability and linkage between these two regions. We added two sets of pacemaker experiments related to the Indian Ocean and four single forcing large ensembles, complementing the definition and effects of external forcing.

1. Title

“Western–central Pacific” is not very clear. Perhaps “Western–central Tropical Pacific” is better.

Response: We thank the reviewer for this suggestion. “Western–central Tropical Pacific” is more concise. We have revised the title.

2. Abstract

L25 "The interbasin linkage generates a broad and profound impact."

Adding a little more background information at the beginning of the abstract (perhaps two sentences) would help readers better understand the implication of the results.

Response: We appreciate the reviewer's suggestion. To address the incomplete information in the first sentence of the abstract, we have included the following sentence: "The impact of interbasin linkage on the weather/climate and ecosystems is significantly broader and profounder than that of only appearing in an individual basin." This addition aims to specify the wide-ranging impacts of the teleconnection between the Indo-Pacific basins.

L29 "external forcing"

In the discussion (L241), the term "the external forcing" is used to signify "anthropogenic forcing". Using a more explicit expression or term related to global warming (e.g., increase of greenhouse gas) would clarify the meaning of "external forcing".

Response: Thank you for your good suggestion. In our revised manuscript, we specified the external forcing as the greenhouse gases and volcanoes, which is important to affect the Tropical Indian Ocean warming (by the warming trend) and decadal variabilities (by the volcanoes) last decades. We further elucidate how "external forcing" impacts the relationship between western Australian coast and the Western-central tropical Pacific. We have added two pacemaker experiments related to Indian Ocean and four single forcing large ensembles:

One pacemaker experiment is TIOGA using the National Center for Atmospheric Research (NCAR) Community Earth System Model version 1 (CESM1), in which the simulated SST in the tropical Indian Ocean and part of the western Pacific warm pool region is restored to observed SST from Extended Reconstructed SST version 3b (ERSSTv3b), with fully restored SST anomalies (SSTA) for 15°N-15°S, from the African Coast to 161°E, and buffer zones at 15°N-20°N, 15°S-20°S, and 161°E-180° (Hurrell et al., 2013; Zhang et al., 2019; Yang et al., 2020). This TIOGA demonstrates the influence of the tropical Indian Ocean and Western warm pool.

Another pacemaker experiment is IOGA using NCAR CESM1, in which the simulated SST in the Indian Ocean (north of 50°S) is restored to observed SST from ERSSTv3b (Hurrell et al., 2013; Zhang et al., 2021). This IOGA highlights the influence of the Indian Ocean, specifically, the effect of the western Australian coast area on Niño4w and their relationship.

Four single forcing large ensembles (Simpson et al., 2023) are employed to examine the role of external forcings further. They are GHG only (only

greenhouse gases evolving, having 15 members); BMB only (Only biomass burning aerosols evolving, having 15 members); EE (everything else evolving, having 15 members, that is, all forcings are time evolving except GHGs, AAERs including BMB which means greenhouse gases and anthropogenic (including biomass burning) aerosols are held fixed at 1850 level; xAER (Everything time evolving except anthropogenic aerosols, having 10 members).

Results from TIOGA experiments reveal that the changes in tropical Indian Ocean and part of western Pacific can generate significant SST changes in the central-western tropical Pacific and off western Australian coast, accounting for a large fraction of the observed NNI after 1985; Before 1985, however, the simulated NNI has a weaker amplitude and opposes to the observation (Fig. RA1c; Fig. 3c of revised manuscript). The simulated NNI and Niño4w index in TIOGA are out of phase with a correlation coefficient of -0.74 (Fig. RA1d; Fig. 3d of revised manuscript). These results suggest that SSTA in tropical Indian Ocean and western Pacific play an important role in causing the enhanced NN-Niño4w linkage in recent decades. This is because a warm (cold) SSTA in tropical Indian-western Pacific basin drives equatorial easterly (westerly) wind anomaly in the central-western tropical Pacific (Fig. RA2; Fig. S14 of revised manuscript), enhances (reduces) equatorial upwelling and causes cold (warm) SSTA in Niño4w region. The enhanced equatorial easterly (westerly) wind anomaly increases (reduces) the Indonesian Throughflow, strengthening Ningaloo Niño (Niña) events. Meanwhile, the colder (warmer) SSTA in Niño4w region can induce cyclonic (anticyclonic) circulation in the Southeast Indian Ocean, together with local air-sea feedback, further strengthening Ningaloo Niño (Niña) magnitudes.

Before 1985, the warming of the tropical Indian Ocean was primarily influenced by the Interdecadal Pacific Oscillation (IPO). Then, the decadal variability in the Indian Ocean was in phase with the IPO (Dong et al., 2017; Zhang et al., 2018). After 1985, the external forcings dominate the changes in the Indian Ocean (Fig. RA1e; Fig. 3e of revised manuscript). Under this influence, the out of phase decadal relationship between Indian Ocean and Pacific basin appears (Fig. R1Af; Fig. 3f of revised manuscript). The warming in the Indian Ocean and the negative phase of IPO can both lead to easterly wind anomalies and cooling in the Niño4w region, further enhancing the heat advection from the western Pacific, ultimately causing the warming off the western Australian coast area (Fig. RA2b; Fig. S14b of revised manuscript).

Therefore, the "external forcing" referred to here primarily denotes the external forcing causing decadal variability in the tropical Indian Ocean and resulting in an enhanced decadal co-variability between these two regions after 1985. we show that the increase in greenhouse gases (GHGs; Fig. RA3; Fig. 4 of revised manuscript, green line) has significantly contributed to the recent trend of rapid

warming in the Tropical Indian Ocean. External forcings that exclude anthropogenic aerosols (experiment xAER; Fig. RA3; Fig. 4 of revised manuscript, blue line) produce a stronger warming trend, which means anthropogenic aerosols somewhat decrease the warming trend in the Indian Ocean. In addition to a warming trend, Experiment xAER also generates decadal variability in the tropical Indian Ocean SST Fig. RA3; (Fig. 4 of revised manuscript, blue line), mainly owing to the forcings by volcanoes (included in everything else experiment, EE; Fig. RA3; Fig. 4 of revised manuscript, purple line). This finding suggests that external forcings by volcanoes is the main cause for TIO SSTA and result in its out of phase relationship with IPO after 1985.

Figure RA1 Possible influencing factors (IPO and external forcing) for decadal variabilities of NNI. **a** Time series of the 21-year sliding variance of the NNI (black line) and the IPO index (blue line) based on ERSSTv5, in which the red shading identifies the positive phase of the IPO during 1978–1997. **b** Time series of the 21-year sliding variance of interannual NNI (blue line) and decadal NNI (red line). Red shading identifies the positive phase of the IPO during 1978–1997. **c** The 8–16 years band-pass filtered decadal NNI in ERSSTv5 and TIOGA. **d** The 8–16 years band-pass filtered decadal NNI and Niño4w index in TIOGA. **e** The 8 years low-pass filtered tropical Indian ocean SSTA (15°N – 15°S , 50° – 160°E) in observations and the ensemble mean of CSM2-LE, and MPI-LE. Blue shading identifies the period 1960-1985. Red shading identifies the period after 1985. **f** The normalized 8-year low-pass filtered Tropical Indian ocean SSTA and IPO index (detrend) in observations.

Figure RA2 Teleconnections for the decadal SST variability off western Australian coast in DJF in different Indian Ocean pacemaker experiments. Correlation between DJF decadal SSTA and the DJF decadal NNI (shading; vectors for wind anomaly; contours for precipitation anomaly) during **a** 1950–1990 and **b** 1990–2013 in CESM1 tropical Indian ocean (TIOGA) pacemaker experiments. **c** and **d** as in **a** and **b** but for the 200-hPa velocity potential anomaly (shading; vectors for divergent wind; contours for precipitation). Red/blue contours represent positive/negative precipitation significant at the 95% confidence level. Stippled areas are where the regression results are statistically significant at the 95% confidence level. **e**, **f**, **g**, and **h** are for 1950–1990 and 1990–2019 in CESM1 Indian ocean (IOGA) pacemaker experiments.

Figure RA3 The contribution of single external forcing to the tropical Indian Ocean SST.

The 8 years low-pass filtered tropical Indian ocean SSTA (15°N–15°S, 50°–160°E) in ERSSTv5 (black) and the ensemble mean of CESM2-LE (red) and four sets of CESM2 single forcing large ensembles: xAER (blue), GHG (green), BMB and EE (orange), and EE (purple). Light blue shading identifies the period 1960–1985. Light red shading identifies the period after 1985.

Reference:

1. Hurrell JW, et al. The community earth system model: a framework for collaborative research. *Bulletin of the American Meteorological Society* 94, 1339-1360 (2013).
2. Zhang L, et al. Indian Ocean warming trend reduces Pacific warming response to anthropogenic greenhouse gases: An interbasin thermostat mechanism. *Geophysical Research Letters* 46, 10882-10890 (2019).
3. Yang D, et al. Role of tropical variability in driving decadal shifts in the Southern Hemisphere summertime eddy-driven jet. *Journal of Climate* 33, 5445-5463 (2020).
4. Zhang L, Han W. Indian ocean dipole leads to Atlantic Niño. *Nature Communications* 12, 5952 (2021).
5. Zhang L, Han W, Sienz F. Unraveling causes for the changing behavior of the tropical Indian Ocean in the past few decades. *Journal of Climate* 31, 2377-2388 (2018).
6. Dong L, McPhaden MJ. Why has the relationship between Indian and Pacific Ocean decadal variability changed in recent decades? *Journal of climate* 30, 1971-1983 (2017).
7. Zhang L, Han W, Sienz F. Unraveling causes for the changing behavior of the tropical Indian Ocean in the past few decades. *Journal of Climate* 31, 2377-2388 (2018).
8. Simpson IR, et al. The CESM2 single-forcing large ensemble and comparison to CESM1: Implications for experimental design. *Journal of Climate* 36, 5687-5711 (2023).

3. L42 "the western Pacific (including eastern Indian Ocean...."

This sentence is confusing. The western Pacific does not include the eastern Indian Ocean.

Response: We thank the reviewer for this suggestion. We have deleted "the eastern Indian Ocean".

4. L51 "varia"

Should be "variation".

Response: We thank the reviewer for this suggestion. We have corrected this word.

5. L54 "most important"

Important for what? Does this mean "strongest"?

Response: Yes, according to the precious studies, the SST index of western-central tropical Pacific (Niño4w) region has the strong correlation with NNI (Feng et al., 2013; Zhang et al., 2018). We have corrected this sentence to "The western-central Pacific (Niño4w) region is the remote area with strong linear correlation with Ningaloo Niño variability."

Reference:

1. Feng M, et al. Decadal increase in Ningaloo Niño since the late 1990s. *Geophysical Research Letters* 42, 104-112 (2015).
2. Zhang L, Han W. Impact of Ningaloo Niño on tropical Pacific and an interbasin coupling mechanism. *Geophysical Research Letters* 45, 11,300-311,309 (2018).

6. L84 "NNI, which is defined as the area-weighted mean SST anomaly (SSTA) in the wedged area"

This definition is different from the most commonly used NNI (area average SST for 110°-116°E, 22°-32°S or 108°E-116°E, 22°-28°S). Are the results sensitive to the definition used? For example, is STD on the decadal time scale similar when using the commonly used NNI? The justification for the choice of this specific definition could be included in the method section.

Response: We adopted the NNI definition from Zhang (2018). In the regression results, a wedge-shaped sea surface temperature mode was displayed. The wedge-shaped is similar to 'wedge of SST warming' pattern employed the commonly used NNI. Therefore, we opted for the definition of the NNI region based on this wedge-shaped sea surface temperature pattern. We calculated the STD of different chosen region on the decadal time scale as suggested, for the region in our manuscript, the STD is 0.51°C, for the region 110°-116°E, 22°-32°S is 0.51°C, and for the region 108°E-116°E, 22°-28°S is 0.52°C. The above results show the rationality of our region selection. We have added this discussion to method section: "This definition can comprehensively capture the warming pattern off western Australian coast as being wedge-shaped. Compared with the traditional definition (Marshall et al., 2015; Kataoka et al., 2014) whose area-weight-averaged SSTA is based on the region 110°-116°E,

22°-32°S , similar NNI variation and amplitude (denoted as STD) are obtained.”

Reference:

1. Zhang L, Han W, Li Y, Shinoda T. Mechanisms for generation and development of the Ningaloo Niño. *Journal of Climate* 31, 9239-9259 (2018).
2. Marshall AG, Hendon HH, Feng M, Schiller A. Initiation and amplification of the Ningaloo Niño. *Climate Dynamics* 45, 2367-2385 (2015).
3. Kataoka T, Tozuka T, Behera S, Yamagata T. On the Ningaloo Niño/Niña. *Climate dynamics* 43, 1463-1482 (2014).

6. L86. “The standard deviations (STDs) of NNI are 0.92 °C (1950-1985) and 1.07 °C 87 (1985-2020).”

The quality of the SST data could be different in these two periods because of the satellite measurements in recent years. Is the uncertainty due to the SST data quality smaller than the difference in STD between the two periods?

Response: Thank for your comments. Extended Reconstruction Sea Surface Temperature Version 3b (ERSSTv3b; Smith et al., 2008) does not include satellite data. Therefore, we used ERSSTv3b to test the uncertainty due to satellite measurements (Fig. RA4). We calculated the wavelet of NNI and Niño4w and the 21-year sliding correlation between these two regions, and the results are similar with that in ERSSTv5, HadISST1, and IAP. This confirms that the enhanced decadal variability and linkage between these two regions in our study aren't influenced by the satellite measurement data since the 1980s.

Figure RA4 SST variability off western Australian coast and the western-central tropical Pacific and their relationship in ERSSTv3b. **a** The 21-year sliding correlation coefficients between the ensemble mean NNI and the ensemble mean Niño4w index, in which the red line signifies results statistically significant at the 95% confidence level. **b** Wavelet spectrum of the ensemble mean NNI. The red lines mark the periodicity of 8–16 years. **c** Wavelet spectrum of the ensemble mean Niño4w index.

Reference:

1. Smith TM, Reynolds RW, Peterson TC, Lawrimore J. Improvements to NOAA's historical merged land-ocean surface temperature analysis (1880–2006). *Journal of climate* 21, 2283–2296 (2008).

7. L93 “8–16 years”

It is beneficial to state more explicitly in this section that the decadal time scale in this study is “8–16 years”, although it is stated in the method section later.

Response: We thank the reviewer for this suggestion. At the end of the first part of the results, we have emphasized that the decadal timescale in our study is mainly 8–16 years as suggested. We added “In the following, we focus on the decadal timescale with 8–16 years.”

8. L134 “The center of negative anomaly shifts westward”

Before 1985, strong correlations exist in the eastern tropical Pacific. However, this may not necessarily indicate an interbasin linkage between western Australian coast and eastern tropical Pacific during this period. Or is it an interbasin linkage? Please explain.

Response: We thank the reviewer for this suggestion. It represents an interbasin linkage between the western Australian coast and the eastern tropical Pacific. This linkage has experienced a westward shift after 1985, showing the Western–central tropical Pacific the primary area of interaction between the NNI and the tropical Pacific.

9. L145 “eastern”

Should be “easterly”.

Response: We thank the reviewer for correction. We have modified this word.

10. L148 “norther”

Should be “northerly”.

Response: We thank the reviewer for correction. We have modified this word.

11. L151 “Weakening of coastal upwelling”

Also, northerly winds reduce the wind speed and thus evaporative cooling (latent heat flux) decreases.

Response: We thank the reviewer for this crucial comment. Moreover, the WES feedback we previously calculated is not solely focused on the contribution of wind to original latent heat. The $Q_{i\lambda}^{a'}$ includes wind speed, relative humidity, and stability effects, which are not directly tied to Sea Surface Temperature (SST) but are due to atmospheric adjustments (Xie et al., 2010). Therefore, the changes in 'latent heat flux' and the wind might have some inconsistencies.

Reference:

1. Xie S-P, Deser C, Vecchi GA, Ma J, Teng H, Wittenberg AT. Global warming pattern formation: Sea surface temperature and rainfall. *Journal of Climate* 23, 966-986 (2010).

12. L159 "The latent heat flux is largely unchanged between these two periods"
Fig. 2 shows that local winds (and thus wind speed) are largely changed between these two periods. So the latent heat flux should be largely changed as well, which is inconsistent with the results. Please clarify.

Response: We thank the reviewer for this comments. Please see the response to comment 11. We used the $Q_{i\lambda}^{a'}$, which includes wind speed, relative humidity, and stability effects. The changes in 'latent heat flux' and the wind might have some inconsistencies.

12. L166 "northern wind anomaly"
Should be "northerly wind anomaly".

Response: We thank the reviewer for correction. We have modified this word.

13. L179 "centra-eastern"
Should be "central-eastern"

Response: We thank the reviewer for correction. We have modified this word.

14. L205 "centra-eastern"
Should be "central-eastern"

Response: We thank the reviewer for correction. We have modified this word.

15. L223 "This indicates external that"
Correct to "This indicates that external"

Response: We thank the reviewer for correction. We have modified this sentence.

16. L245 "convection in situ"
Not clear what this means.

Response: We thank the reviewer for correction. We modified this to "in situ convection" to show that the convection is also over the Maritime Continent

and western Pacific.

17. L255 "northern surface wind anomaly"

Should be "northerly surface wind anomaly"

Response: We thank the reviewer for correction. We have modified this word.

18. L279 "eastern wind anomaly"

Should be "easterly wind anomaly"

Response: We thank the reviewer for correction. We have modified this word.

We also corrected all the similar grammar mistakes.

19. L332 "Japanese 55-year Reanalysis (JRA55)"

The time period covered in OAFlux is similar. Is the decadal variability in surface flux data from JRA55 consistent with OAFlux? Also, the uncertainty of surface flux data in this region should be mentioned (e.g., Feng and Shinoda 2019).

Response: Thank you for your valuable comments. The uncertainty between different datasets is important for this study. According to previous studies (Feng and Shinoda 2019), especially during the boreal winter when the Ningaloo Niño develops, the results of surface heat flux from different datasets vary greatly, which poses certain difficulties to our qualitative research. We have added it in discussion: "In heat budget analysis, the surface heat fluxes vary substantially with the datasets, especially during the boreal winter when the Ningaloo Niño develops (Feng and Shinoda 2019). However, the increased heat advection is supported by different datasets and methods."

To further validate the importance of oceanic dynamic, we did a composite analysis of the mixed-layer heat budget off western Australian coast (in Ningaloo Niño region, (Table RA1) during the development phase of decadal Ningaloo Niño events. In this phase, Ningaloo Niño region experiences significant warming for both periods, but the warming rate is larger for the post-1985 period (Table RA1), consistent with the intensified SSTA. Increased ocean heat advection plays a dominant role in causing the intensified SST variability. As shown in Table RA1, the largest terms of the heat budget are the surface heat flux and the residue. Here the residue term represents the vertical heat transport due to upwelling of subsurface water into the mixed layer and all other non-resolved processes. The net effect of the horizontal advection terms is to diverge the heat away from the Ningaloo Niño region, and this effect is larger before 1985 than after 1985. Combined with residue term, our budget analysis suggests that the reduced heat divergence due to horizontal advection and the reduced upwelling after 1985 lead to an enhanced Ningaloo Niño amplitude on decadal timescale. Meridional heat advection primarily originates from the western tropical Pacific, driving more warm water towards the northwestern Australian coast. With the strengthening of the cyclonic wind anomaly, the resulting eastward geostrophic current anomalies carrying

warmer water have led to an overall increase in SST off western Australian coast.

The equation for mixed-layer heat budget follows:

$$\frac{\partial T'}{\partial t} = -\left(\bar{u} \frac{\partial T'}{\partial x} + u' \frac{\partial \bar{T}}{\partial x} + u' \frac{\partial T'}{\partial x}\right) - \left(\bar{v} \frac{\partial T'}{\partial y} + v' \frac{\partial \bar{T}}{\partial y} + v' \frac{\partial T'}{\partial y}\right) + \frac{Q_{net'}}{\rho_0 C_p h} + R'$$

T , u , and v represent the oceanic temperature, zonal and meridional current velocities averaged in the mixed-layer depth, respectively. $\frac{\partial T'}{\partial t}$ is temperature

Tendency. A variable (such as T , u , and v) can be decomposed into the sum of the climatological mean state (e.g. \bar{u}) and anomaly state (e.g. u'). Q_{net}' is the "net heat flux", which accounts for the sum of shortwave radiation, longwave radiation, latent heat fluxes, and sensible heat fluxes. ρ_0 indicates sea water density (1026 kg/m^3) and C_p is the ocean heat capacity ($3986 \text{ J/(kg} \cdot \text{K)}$), h is mixed-layer depth. R' is the "residual" term, which accounts for the remaining heat budget components not explicitly resolved by the used data in the equation. The results presented above were obtained through the vertical integration of the heat budget across the mixed layer.

Table RA1 The average of mixed-layer heat budget terms of Ningaloo Niño region by choosing decadal Ningaloo Niño events (before 1985 and after 1985). The chosen events are December 1970 to December 1972, and January 1981 to August 1983 (before 1985); June 1986 to June 1988, and October 1995 to August 1997 (after 1985), monthly data after 8 - 16 years band-pass filter is used.

Average of decadal Ningaloo Niño events (10^{-9} K/s)	Before 1985	After 1985
$\frac{\partial T'}{\partial t}$	1.44	1.86
$\frac{Q_{net}'}{\rho_0 C_p h}$	-12.88	-9.16
R'	27.37	15.26
$u' \frac{\partial T'}{\partial x}$	0.77	0.51
$u' \frac{\partial \bar{T}}{\partial x}$	0.16	2.23
$\bar{u} \frac{\partial T'}{\partial x}$	-1.51	0.64
$v' \frac{\partial T'}{\partial y}$	0.79	-0.74
$v' \frac{\partial \bar{T}}{\partial y}$	-9.94	-7.05

$\bar{v} \frac{\partial T'}{\partial y}$	-6.08	1.21
--	-------	------

Reference:

1. Feng X, Shinoda T. Air-sea heat flux variability in the southeast Indian Ocean and its relation with Ningaloo Niño. *Frontiers in Marine Science* 6, 266 (2019).

20. Reference section

Most of the references lack journal names.

Response: Thank you for your comments. We have revised all the references.

21. L592 Data and materials availability

Availability of two pacemaker simulations is not provided.

Response: Thank you for your comments. We have added the data availability for Tropical Indian Ocean pacemaker experiment. We have Data from some pacemaker experiments (CESM1 Pacific pacemaker experiment, FGOALS-f3 pacemaker experiment, and Indian Ocean pacemaker experiment): "The related data of some pacemaker experiments (CESM1 Pacific pacemaker experiment, FGOALS-f3 Pacific pacemaker experiment, and Indian Ocean pacemaker experiment) are available via http://data.lasg.ac.cn/lpf/data_availability-POGA-IOGA.zip."

Reviewer #2 (Remarks to the Author):

Review of the paper entitled "Emergence of Decadal Linkage between Western Australian Coast and Western-central Pacific" by Dr Lin and colleagues.

Main

This work investigates the recent enhancement of decadal variability observed in the northwest Australian coast. Through observational data analyses and model experiments, the paper suggests that the anomalous warming of the western Pacific tropical Ocean contributes to enhanced variability via atmospheric teleconnections. The Paper also suggests an emergence of decadal linkage between the two regions under the western Pacific warming which is likely to persist in the future given the projected warming of the tropical Pacific. The role of western Pacific variability on the development of Ningaloo Niño (NN) has been reported earlier. Feng et al. (2015) also suggested that anthropogenic warming in the western Pacific may have an additional role in modulating the NN at decadal time scales. This paper takes this idea further with detailed research to show that recent decadal variations of NN might have an

independent existence from IPO under strong external warming, leading to a 'new decadal link' between the two regions. This is an exciting finding!

I appreciate the enthusiasm of the authors to tackle a complex research problem that has narrow lines separating the role of anthropogenic warming and natural variations for controlling variability over different time scales. Given the complexities of the problem, I still have quite a few concerns and questions which I list below. I hope clearing these questions will bring some more clarity to the presentation of the paper.

Response: We would like to thank the reviewer for the constructive comments/suggestions. We have significant modifications based on the reviewers' comments, adding two sets of pacemaker experiments related to the Indian Ocean and four single forcing large ensembles to explore the specific effects of external forcing. We also applied the 8–16 years band-pass filtering to both the NNI and variable fields to ensure our study focusing on the decadal timescale, and we discussed the linkage on an interannual scale. To verify the role of oceanic dynamic, we added calculations of the mixed-layer heat budget. We also added some important discussion.

Major comments:

1) What is the rationale behind using EEMD to obtain components of variability over different time scales? For instance, how do the authors make sure that the decadal component (fifth and sixth components combined) of indices has variability only above 7 years? Also, it seems that only indices are filtered for different time scales but not the variables (SST, wind, etc.). You may get spurious regression estimates if the dependent (SST for example) and independent (indices) operate over different time scales. I would like the authors to make a little more serious effort to clarify these aspects in the paper.

Response: EEMD, through adaptive decomposing steps, decomposes the time series into oscillatory components at different time scales. EEMD is an improved version of Empirical Mode Decomposition (EMD). In comparison to EMD, EEMD addresses the 'mode mixing' issue, resulting in a more stable decomposition outcome. EMD represents data through a series of filtering steps, expressing it as a combination of a finite number of periodic components (IMFs) and a trend term. This process involves connecting local maxima and minima of the original data or the series obtained by subtracting several oscillatory components using cubic spline functions to form upper and lower envelopes. The original data is then subtracted from these two envelopes' local means, and this process is repeated until the local mean is zero everywhere, yielding the oscillatory component for a specific period. EEMD, on the other hand, leverages the principle of multiple measurements averaging by introducing appropriately

sized white noise into the original data to simulate scenarios of multiple observations. Through multiple calculations and ensemble averaging, the final result is obtained.

To ensure that the decadal component includes variability only above 7 years, the EEMD process involves decomposing the time series into different IMFs, each representing a specific frequency or timescale. We used power spectra to calculate the peak for each IMFs, thus determining their timescales variations (Fig. RB1). By combining the fifth and sixth IMFs, which correspond to timescales longer than 7 years, we isolated and obtained the decadal variability. We also used 8–16 year band-pass filter in revised manuscript.

We have provided correlation maps of the decadal NNI and the decadal variable fields (SST, surface wind, etc.), utilizing an 8–16 years bandpass filter for the decadal analysis. The results show that the positive SST anomaly mode still occurs in the Ningaloo Niño region after 1985. Meanwhile, in the tropical Pacific, the strong negative correlation areas have slightly shifted westward, and the stronger easternly wind over the western - central tropical Pacific (Niño4w region; Fig. RB2, red box). The connection between the decadal NNI and Niño4w has strengthened after 1985 on decadal timescale.

Figure RB1 Power spectra of the EEMD components using different datasets, with the first column for ERSSTv5, the second column for HadISST, and the third column for IAP.

Figure RB2 Correlation between the decadal DJF SSTA (shading; vectors: surface wind) and the decadal DJF NNI based on ERSSTv5 and HadISST. The blue/red box represents the Ningaloo Niño and the Niño4w regions, respectively. Stippled areas denote where the regression results are statistically significant at the 95% confidence level.

2) (Page 17) Analysis of SST pattern: This section doesn't speak of ocean heat transport (D) well. The balance between D and Q is set as an assumption but hasn't been tested explicitly. In fact, previous works on low-frequency NN and its mechanisms (Feng et al. 2015) showed that heat advection from the western Pacific plays an important role. The present paper summarises the WES and ocean transport contributions in the form of regression coefficients (table S2) which is very hard to understand. I do not even understand what are the two sets of values presented for each period in this table. I find that the way of treatment of ocean heat transport is an important drawback of this study. In fact, there is no description of how and where heat transport anomalies are calculated. Atmospheric teleconnections might be playing an important role as discussed in this paper, but I would like to see if the role of oceanic heat transport can be expelled more undoubtedly. For instance, there might be a reason why ITF doesn't contribute much to the decadal linkage, however, this study doesn't explore that.

Response: We thank the reviewer for this remarkable comment. In our previous study, we worked under the assumption that ocean heat transport and net surface heat flux are balanced against each other over long timescales, meaning that D'_0 is essentially represented by the negative of Q_{net} . In addition, we calculated the contributions of different items in the Ningaloo Niño region: latent heat representing atmospheric forcing, shortwave, longwave, and sensible heat. In the previous manuscript, the periods represented inside and outside the parentheses were not clearly expressed; we have now corrected this: outside the brackets represents the decadal contributions, while inside brackets the contributions from the raw data. Our findings agree what you mentioned: changes in ocean heat transport are important.

To further validate the importance of oceanic dynamic, we did a composite analysis of the mixed-layer heat budget off western Australian coast (in Ningaloo Niño region, (Table RB1) during the development phase of decadal Ningaloo Niño events. In this phase, Ningaloo Niño region experiences significant warming for both periods, but the warming rate is larger for the post-1985 period (Table RB1), consistent with the intensified SSTA. Increased ocean heat advection plays a dominant role in causing the intensified SST variability. As shown in Table RB1, the largest terms of the heat budget are the surface heat flux and the residue. Here the residue term represents the vertical heat transport due to upwelling of subsurface water into the mixed layer and all other non-resolved processes. The net effect of the horizontal advection terms is to diverge the heat away from the Ningaloo Niño region, and this effect is larger before 1985 than after 1985. Combined with residue term, our budget analysis suggests that the reduced heat divergence due to horizontal advection and the reduced upwelling after 1985 lead to an enhanced Ningaloo Niño amplitude on decadal timescale. Meridional heat advection primarily originates from the western tropical Pacific, driving more warm water towards the northwestern Australian coast. With the strengthening of the cyclonic wind anomaly, the resulting eastward geostrophic current anomalies carrying warmer water have led to an overall increase in SST off western Australian coast. The results are consistent with the diagnosis that the ocean dynamical role is important (Table S1 of original manuscript) and the findings of other researchers (Feng et al., 2015).

The equation for mixed-layer heat budget follows:

$$\frac{\partial T'}{\partial t} = - \left(\bar{u} \frac{\partial T'}{\partial x} + u' \frac{\partial \bar{T}}{\partial x} + u' \frac{\partial T'}{\partial x} \right) - \left(\bar{v} \frac{\partial T'}{\partial y} + v' \frac{\partial \bar{T}}{\partial y} + v' \frac{\partial T'}{\partial y} \right) + \frac{Q_{net'}}{\rho_0 C_p h} + R'$$

T, u, and v represent the oceanic temperature, zonal and meridional current velocities averaged in the mixed-layer depth, respectively. $\frac{\partial T'}{\partial t}$ is temperature

Tendency. A variable (such as T, u, and v) can be decomposed into the sum of the climatological mean state (e.g. \bar{u}) and anomaly state (e.g. u'). $Q_{net'}$ is the "net heat flux", which accounts for the sum of shortwave radiation, longwave radiation, latent heat fluxes, and sensible heat fluxes. ρ_0 indicates sea water density (1026 kg/m^3) and C_p is the ocean heat capacity ($3986 \text{ J/(kg} \cdot \text{K)}$), h is mixed-layer depth. R' is the "residual" term, which accounts for the remaining heat budget components not explicitly resolved by the used data in the equation. The results presented above were obtained through the vertical integration of the heat budget across the mixed layer.

Table RB1 The average of mixed-layer heat budget terms of Ningaloo Niño region by choosing decadal Ningaloo Niño events (before 1985 and after 1985). The chosen events are December 1970 to December 1972, and January 1981 to August 1983 (before 1985); June 1986 to June 1988, and

October 1995 to August 1997 (after 1985), monthly data after 8 - 16 years band-pass filter is used.

Average of decadal Ningaloo Niño events (10^{-9} K/s)	Before 1985	After 1985
$\frac{\partial T'}{\partial t}$	1.44	1.86
$\frac{Q_{net}'}{\rho_0 C_p h}$	-12.88	-9.16
R'	27.37	15.26
$u' \frac{\partial T'}{\partial x}$	0.77	0.51
$u' \frac{\partial \bar{T}}{\partial x}$	0.16	2.23
$\bar{u} \frac{\partial T'}{\partial x}$	-1.51	0.64
$v' \frac{\partial T'}{\partial y}$	0.79	-0.74
$v' \frac{\partial \bar{T}}{\partial y}$	-9.94	-7.05
$\bar{v} \frac{\partial T'}{\partial y}$	-6.08	1.21

Reference:

1. Feng M, et al. Decadal increase in Ningaloo Niño since the late 1990s. *Geophysical Research Letters* 42, 104-112 (2015).

3) Related to the previous comment: The role of (slow) upper ocean heat content variations in the western Pacific and its transmission to the West Australian coast (WAC), affecting SST and sea level there, is indeed discussed earlier (Feng et al. 2015). This paper almost fully neglects the role of ocean waveguides in shaping the low-frequency SST variations in the WAC. I was wondering how tide-gauge sea-level measurements at Fremantle which is identified to be a good indicator of heat content variations in the WAC (Feng et al. 2003, 2015) would complement the overall result of this paper. Even though the authors try to establish the absence of oceanic teleconnection (S3?), the description of figure 2 (Page 7) uses the ocean transport (Leeuwin current) to explain the spatial structures. Do authors believe that the influence of Leeuwin current is confined only to the NN region?

Response: We thank the reviewer for such a insightful comment. As we stated and obtained in **Table RB1**, the reduced heat divergence due to horizontal advection and the reduced upwelling after 1985 lead to an enhanced Ningaloo Niño amplitude on decadal timescale. The results show that, both in terms of the previous SST pattern analysis and the mixed layer heat budget calculation,

the importance of heat advection is confirmed, just as Feng previously mentioned. The heat advection north side of the NNI region (including western Pacific) is significant. This leads to an intensification of the Leeuwin Current, ultimately causing an increase in sea temperatures off western Australian coast. Also, Due to the cyclonic anomaly off western Australian coast, the induced eastward geostrophic current anomalies have brought more warm water to the coast. Therefore, we believe that the role of the Leeuwin Current is not limited only to the Ningaloo Niño region.

4) There is a systematic difference between the NNI-regressed spatial SST maps in Figure 1 and Figure 3. There is no variability seen in the western Pacific in figure 1 (observation) but there are strong positive SST anomalies in the western Pacific in figure 3 (coupled model). Can the authors elaborate on this difference further? An intercomparison of figures 1, 3, and S8 prompts me to think that the NN in coupled models is tightly linked to canonical ENSO and affected by its interannual variations but this is not the case *for observation*. This may have strong implications for the overall result presented in this paper. For instance, in the pacemaker experiments, the ENSO variations dominate and the effect of Maritime continent warming after 1990 might be nil or weak. But, in observations, Maritime warming might play an important role in shaping the decadal linkage. I would ask the authors to redo figure S5 but for models (pacemaker experiments) for the two periods to see this further.

Response: Thank you very much for your valuable comments, which are of great importance for enhancing the integrity of the conclusions in our manuscript. Firstly, we recognize the existence of enhanced coherence of interannual SSTA between NNI and Niño4w after 1985, such as in Fig. S4 of original manuscript where IAP SST seems an outlier for capturing the 4-6yr peak, compared to the period before 1985. Additionally, in previous Fig 3b, we also demonstrated the growth of NNI on interannual scale. La Niña can also promote a stronger Leeuwin Current (Feng et al., 2015). Therefore, as a crucial interannual signal in the tropical Pacific, the role of ENSO in the linkage between the two regions on interannual scale cannot be underestimated. However, the focus of this manuscript is on decadal timescale. In light of your comments and suggestions, we have made significant modifications and enhancements to section 3, including the addition of two sets of pacemaker experiments related to Indian Ocean to explore the sources of decadal scale fluctuations and linkages between the NNI and Niño4w regions, and four single forcing large ensembles to study the role of external forcing.

Firstly, we refined our filtering and ensemble mean methods. We now apply an 8–16 years bandpass filter to the NNI and variable fields in both observation and pacemaker experiments. This will ensure the analysis is on the decadal timescale. In the CESM1.2 Pacific pacemaker, correlation maps before and after

1985 show enhanced warming along the western Australian coast and an intensified negative pattern of the IPO. We also added correlation maps similar to Fig S5 (Fig. RB3; Fig. S11 of revised manuscript), indicating that the IPO indeed contributes to the linkage between the NNI and Niño4w regions on decadal timescale. From observations, it is evident there are strong positive SST anomalies in the western Pacific now (Fig. RB2 b and d).

Therefore, we introduce two new pacemaker experiments: TIOGA using the National Center for Atmospheric Research (NCAR) Community Earth System Model version 1 (CESM1), in which the simulated SST in the tropical Indian Ocean and part of the western Pacific warm pool region is restored to observed SST from Extended Reconstructed SST version 3b (ERSSTv3b), with fully restored SST anomalies (SSTA) for 15°N-15°S, from the African Coast to 161°E, and buffer zones at 15°N-20°N, 15°S-20°S, and 161°E-180° (Hurrell et al., 2013; Zhang et al., 2019; Yang et al., 2020). Another experiment is IOGA using NCAR CESM1, in which the simulated SST in the Indian Ocean (north of 50°S) is restored to observed SST from ERSSTv3b (Hurrell et al., 2013; Zhang and Han, 2021).

Results from TIOGA experiments reveal that the changes in tropical Indian Ocean and part of western Pacific can generate significant SST changes in the central-western tropical Pacific and off western Australian coast, accounting for a large fraction of the observed NNI after 1985; Before 1985, however, the simulated NNI has a weaker amplitude and opposes to the observation (Fig. RB4c; Fig. 3c of revised manuscript). These results suggest that SSTA in tropical Indian Ocean and western Pacific play an important role in causing the enhanced NN-Niño4w linkage in recent decades. This is because a warm (cold) SSTA in tropical Indian-western Pacific basin drives equatorial easterly (westerly) wind anomaly in the central-western tropical Pacific (Fig. RB5; Fig. S14 of revised manuscript), enhances (reduces) equatorial upwelling and causes cold (warm) SSTA in Niño4w region. The enhanced equatorial easterly (westerly) wind anomaly increases (reduces) the Indonesian Throughflow, strengthening Ningaloo Niño (Niña) events. Meanwhile, the colder (warmer) SSTA in Niño4w region can induce cyclonic (anticyclonic) circulation in the Southeast Indian Ocean, together with local air-sea feedback, further strengthening Ningaloo Niño (Niña) magnitudes.

From ensemble mean results from CESM2-LE and MPI-LE, it can be seen that before 1985, the TIO SSTA did not exhibit strong decadal fluctuations under the influence of external forcing. However, after 1985, the TIO SST decadal fluctuations in large ensembles and observations agree well (Fig. RB4e; Fig. 3e of revised manuscript), indicating that the decadal fluctuations in TIO after 1985 are primarily controlled by external forcing, consistent with previous studies (Dong et al., 2017; Zhang et al., 2018). Initially, TIO SST was under

the control of the IPO, but subsequently, external forcing became the dominant influence, resulting in decadal fluctuations in TIO SST (Dong et al., 2017; Zhang et al., 2018). In contrast, IPO, after 1985, does not resemble observations (Fig. RB6c; Fig. S15c of revised manuscript).

The IOGA highlights the influence of the Indian Ocean, specifically, the effect of the western Australian coast area on Niño4w region and their relationship. Our results capture an enhanced decadal linkage between NNI and Niño4w after 1990 (Fig. RB5f; Fig. S14f of revised manuscript), indicating that the NNI region can also influence the Niño4w region.

In the TIOGA experiments, the NNI and Niño4w index consistently shows negative correlation before and after 1985 (Fig. RB4d; Fig. 3d of revised manuscript), whereas the simulated NNI agrees with observed one only after 1985 (Fig. RB4c; Fig. 3c of revised manuscript). This is because before 1985, the Niño4w SSTA generated by Indian Ocean SST is out of phase with that of the IPO, whereas after 1985 they are generally in phase (Fig. RB5b; Fig. S15b of revised manuscript). The warm (cold) Indian Ocean SSTA and negative (positive) IPO act in concert to intensify Niño4w SSTA and thus NNI after 1985, while the Indian Ocean SSTA weakens the IPO effect before 1985 ((Fig. RB4f; Fig. 3 f of revised manuscript), causing the enhanced decadal linkage between NNI and Niño4w index after 1985 as shown by observations. Thus, the decadal linkage between NNI and Niño4w has been explained.

Four single forcing large ensembles (Simpson et al., 2023) are employed to examine the role of external forcings further. They are GHG (only greenhouse gases evolving, having 15 members); BMB (Only biomass burning aerosols evolving, having 15 members); EE (everything else evolving, having 15 members, that is, all forcings other than those that are time evolving in GHG, AAER or BMB are time evolving. Greenhouse gases and anthropogenic and biomass burning aerosols are held fixed); xAER (Everything time evolving except anthropogenic aerosols, having 10 members).

The above results all stem from the fact that, after 1985, external forcing caused TIO SST to be out of phase with the IPO, leading to the warming trend and decadal fluctuation. Upon examining the impacts of various single external forcings, we show that the increase in greenhouse gases (GHGs; Fig. RB7; Fig. 4 of revised manuscript, green line) has significantly contributed to the recent trend of rapid warming in the Tropical Indian Ocean. External forcings that exclude anthropogenic aerosols (experiment xAER; Fig. RB7; Fig. 4 of revised manuscript, blue line) produce a stronger warming trend, which means anthropogenic aerosols somewhat decrease the warming trend in the Indian Ocean. In addition to a warming trend, Experiment xAER also generates decadal variability in the tropical Indian Ocean SST Fig. RB7; (Fig. 4 of revised

manuscript, blue line), mainly owing to the forcings by volcanoes (included in everything else experiment, EE; Fig. RB7; Fig. 4 of revised manuscript, purple line). This finding suggests that external forcings by volcanoes is the main cause for TIO SSTA and result in its out of phase relationship with IPO after 1985.

Figure RB3 Teleconnections for the decadal SST variability off western Australian coast in DJF in Pacific pacemaker experiments. Correlation between the DJF decadal SSTA and the DJF decadal NNI (shading; vectors for wind anomaly; contours for precipitation anomaly) during a 1950–1990 and b 1990–2014 in CESM1.2 (Pacific pacemaker experiments). c and d as in a and b but for the 200-hPa velocity potential anomaly (shading; vectors for divergent wind; contours for precipitation). Red/blue contours represent positive/negative precipitation significant at the 95% confidence level. Stippled areas are where the regression results are statistically significant at the 95% confidence level. e, f, g, and h are for 1950–1985 and 1985–2014 in FGOALS-f3-L (Pacific pacemaker experiments).

Figure RB4 Possible influencing factors (IPO and external forcing) for decadal variabilities of NNI. **a** Time series of the 21-year sliding variance of the NNI (black line) and the IPO index (blue line) based on ERSSTv5, in which the red shading identifies the positive phase of the IPO during 1978–1997. **b** Time series of the 21-year sliding variance of interannual NNI (blue line) and decadal NNI (red line). Red shading identifies the positive phase of the IPO during 1978–1997. **c** The 8–16 years band-pass filtered decadal NNI in ERSSTv5 and TIOGA. **d** The 8–16 years band-pass filtered decadal NNI and Niño4w index in TIOGA. **e** The 8 years low-pass filtered tropical Indian ocean SSTA (15°N–15°S, 50°–160°E) in observations and the ensemble mean of CESM2-LE, and MPI-LE. Blue shading identifies the period 1960-1985. Red shading identifies the period after 1985. **f** The normalized 8-year low-pass filtered Tropical Indian ocean SSTA and IPO index (detrend) in observations.

Figure RB5 Teleconnections for the decadal SST variability off western Australian coast in DJF in different Indian Ocean pacemaker experiments. Correlation between the DJF decadal SSTA and the DJF decadal NNI (shading; vectors for wind anomaly; contours for precipitation anomaly) during **a** 1950–1990 and **b** 1990–2013 in CESM1 tropical Indian ocean (TIOGA) pacemaker experiments. **c** and **d** as in **a** and **b** but for the 200-hPa velocity potential anomaly (shading; vectors for divergent wind; contours for precipitation). Red/blue contours represent positive/negative precipitation significant at the 95% confidence level. Stippled areas are where the regression results are statistically significant at the 95% confidence level. **e**, **f**, **g**, and **h** are for 1950–1990 and 1990–2019 in CESM1 Indian ocean (IOGA) pacemaker experiments.

Figure RB6 a The 8–16 years band-pass filtered decadal NNI in ERSSTv5 and IOGA. **b** The 8–16 years band-pass filtered decadal Niño4w index in ERSSTv5 and IOGA. **c** The 8 years low-pass filtered IPO index (15°N – 15°S , 50° – 160°E) in observations and the ensemble mean of CESM2-LE, and MPI-LE. Blue shading identifies the period 1960–1985. Red shading identifies the period after 1985.

Figure RB7 The contribution of single external forcing to the tropical Indian Ocean SST.

The 8 years low-pass filtered tropical Indian ocean SST (15°N – 15°S , 50° – 160°E) in ERSSTv5 (black) and the ensemble mean of CESM2-LE (red) and four sets of CESM2 single forcing large ensembles: xAER (blue), GHG (green), BMB+EE (orange), and EE (purple).

and EE (orange), and EE (purple). Light blue shading identifies the period 1960-1985. Light red shading identifies the period after 1985.

Reference:

1. Hurrell JW, et al. The community earth system model: a framework for collaborative research. *Bulletin of the American Meteorological Society* 94, 1339-1360 (2013).
2. Zhang L, et al. Indian Ocean warming trend reduces Pacific warming response to anthropogenic greenhouse gases: An interbasin thermostat mechanism. *Geophysical Research Letters* 46, 10882-10890 (2019).
3. Yang D, et al. Role of tropical variability in driving decadal shifts in the Southern Hemisphere summertime eddy-driven jet. *Journal of Climate* 33, 5445-5463 (2020).
4. Zhang L, Han W. Indian ocean dipole leads to Atlantic Niño. *Nature Communications* 12, 5952 (2021).
5. Dong L, McPhaden MJ. Why has the relationship between Indian and Pacific Ocean decadal variability changed in recent decades? *Journal of climate* 30, 1971-1983 (2017).
6. Zhang L, Han W, Sienz F. Unraveling causes for the changing behavior of the tropical Indian Ocean in the past few decades. *Journal of Climate* 31, 2377-2388 (2018).
7. Feng M, et al. Decadal increase in Ningaloo Niño since the late 1990s. *Geophysical Research Letters* 42, 104-112 (2015).
8. Simpson IR, et al. The CESM2 single-forcing large ensemble and comparison to CESM1: Implications for experimental design. *Journal of Climate* 36, 5687-5711 (2023).

5) Related to the previous comment - (Page 10) authors say 'surprisingly, the nni-Niño4w decadal linkage is found in the ensemble mean of members whose IPO phases are totally opposite to observed phase' – this may simply indicate that, in the ensemble mean, it is not IPO but interannual variability in the equatorial Pacific affects the linkage. In fact, Fig. S11 suggests that there is virtually no decadal power in the ensemble mean Niño4w for R-, indicating that the spatial pattern seen in figure 3f comes from interannual variability. This is contradictory to the argument made, and I would ask the authors to make a revision accordingly either for the analysis or the text at the beginning of Page 11. Also, I would like to ask authors to show spatial maps similar to Figures 3e, f but for the entire 100-member ensemble mean SST (not for R+/- groups).

Response: Thank you for the reviewer's valuable comments. The chosen R- and R+ cannot split the role of IPO and rapid warming. So, in the revision, we do not employ the figures based on the R- and R+. To explain this, two sets of pacemaker experiments related to Indian Ocean are adding to explore the sources of decadal scale fluctuations and linkages between the NNI and Niño4w regions, and we use 8 - 16 years band-pass filter for both index and variable

fields to focus on decadal timescale. Four single forcing large ensembles to study the role of external forcing. We believe these experiments can explain the role of rapid warming, IPO and external forcings in modulating decadal linkage between the NNI and Niño4w index.

We also show the mean SST map and the SST bias between ERSSTv5 and CESM2-LE (CESM2-LE minus ERSSTv5). The warmer tropical Pacific Ocean in CESM2-LE indicate that it fails to capture observed La Niña-like cooling in the eastern Pacific (Fig. RB8).

Figure RB8 **a** The mean state of SST in CESM2-LE ensemble mean. **b** The mean state of SST in ERSSTv5. **c** The bias between the CESM2-LE ensemble mean and ERSSTv5 (CESM2-LE ensemble mean minus ERSSTv5).

6) Figure 3: Do the wavelets represent the ensemble mean spectrum or the spectrum of the ensemble mean index? I think these two are different. Also, there is a 4-year periodicity in the Niño4w in figure 3d which is absent in figure 1e. Shouldn't the ensemble mean spectrum eliminate such interannual signals? Can authors explain why this is so?

Response: We appreciate your comment. We initially calculated the wavelet for each ensemble member and then average them in original manuscript. This approach is chosen because the spectrum of the ensemble mean index tends

to filter out decadal-scale variations. To ensure that the presented results are entirely generated by external forcing, we have corrected the results. Now, all related results are first subjected to ensemble mean before further processing (such as wavelet analysis).

Other comments:

1) Page 4; line 71: Do we have enough observations before 1980 in the western Australian coast to confidently state a post-1980 change? I would like to hear from the authors if they think the enhancement of decadal variations since 1980 is because we have robust measurements since 1980 (satellite) in this region.

Response: Thank for your comments. Extended Reconstruction Sea Surface Temperature Version 3b (ERSSTv3b; Smith et al., 2008) does not include satellite data due to a cold bias in the satellite-derived SSTs that proved difficult to correct. Therefore, we used ERSSTv3b to test the uncertainty due to satellite measurements (Fig. RB9). We calculated the wavelet of NNI and Niño4w and the 21-year sliding correlation between these two regions, and the results is similar with that in ERSSTv5, HadISST1, and IAP. This confirms that the enhanced decadal variability and linkage between these two regions don't be influenced by uncertainty due to the SST data quality such as satellite measurements since the 1980s.

Figure RB9 SST variability off western Australian coast and the tropical

western–central tropical Pacific and their relationship in ERSSTv3b. **a** The 21-year sliding correlation coefficients between the ensemble mean NNI and the ensemble mean Niño4w index, in which the red line signifies results statistically significant at the 95% confidence level. **b** Wavelet spectrum of the ensemble mean NNI. The red lines mark the periodicity of 8–16 years. **c** Wavelet spectrum of the ensemble mean Niño4w index.

Reference:

1. Smith TM, Reynolds RW, Peterson TC, Lawrimore J. Improvements to NOAA's historical merged land–ocean surface temperature analysis (1880–2006). *Journal of climate* 21, 2283-2296 (2008).

2) Statistical significance of the coherence over a 21-year period and other regression estimates. The methodology says the test employs a student's t-test but doesn't explicitly state how the effective sample size (ESS) is estimated. Since the signals are smoothed over decadal periods, the autocorrelation of the indices/time series would reduce the independent samples, and significance tests should consider the ESS.

Response: Thank you for providing insightful feedback. To evaluate statistical significance, we applied a two-tailed Student's t-test. Yes, it's worth noting that our methodology involves low-pass filtering and similar techniques that smooth data over decadal periods, leading to a reduction in degrees of freedom. Consequently, we address this by employing effective degrees of freedom in our analysis, calculated using the following formula (Li et al., 2022):

$$\frac{1}{N_{eff}} \approx \frac{1}{N} + \frac{2}{N} \sum_{j=1}^{N-2} \frac{N-j}{N} \rho_{XX}(j) \rho_{YY}(j)$$

where N is the sample size, $\rho_{XX}(j)$ and $\rho_{YY}(j)$ are the autocorrelations of two time series X and Y at lag.

We have added relevant details in the revised version of the manuscript.

Reference:

1. Li J, et al. Influence of the NAO on wintertime surface air temperature over East Asia: multidecadal variability and decadal prediction. *Advances in Atmospheric Sciences* 39, 625-642 (2022).

3) The IPO definition doesn't follow the reference (Henley et al. 2015) given. Henley et al. suggested a tripolar index instead of EOF.

Response: Thank you for your comments. For consistency throughout the study, the revised manuscript adopts the tripolar index for the IPO.

4) Introduction:

Page 3; line 45: 'Variations in the interbasin linkage between off western Australian coast and the western–central Pacific, linked by the Indo-Pacific

warm pool and the overlying atmospheric circulation, can generate a profound impact on the local and remote climate’.

‘Better rephrase to ‘can impact local and remote climate’.

Response: Thank you for this insightful comment. We have revised this sentence.

5) Page 3; line 50: Not discussing ‘local’ processes well: How much variance is explained by local and remote processes respectively? Similarly, what is the main mode of low-frequency (interannual and above) SST variability on the West Australian coast? Is it Ningaloo Niño or warming associated with the Leeuwin current?

Response: Thank you for this valuable comment., We use Empirical Orthogonal Function (EOF) analysis to identify and understand the patterns and drivers of Ningaloo Niño events, and the EOF1 pattern generally explains approximately 50 % of the total variance (Kataoka et al. 2014; Doi et al. 2013). Based on our results and previous study (Feng et al. 2015), the main mode for both interannual and decadal variability is the wedge-shaped warming pattern along the West Australian coast. The intensification of the Leeuwin Current is crucial for the warming anomalies along the West Australian coast. The main mode of low-frequency (interannual and above) SST variability on the West Australian coast includes both Ningaloo Niño events and warming associated with the Leeuwin Current. We have added “Meanwhile, the colder (warmer) SSTA in Niño4w region can induce cyclonic (anticyclonic) circulation in the Southeast Indian Ocean, together with local air-sea feedback, further strengthening Ningaloo Niño (Niña) magnitudes.”

Reference:

1. Feng M, et al. Decadal increase in Ningaloo Niño since the late 1990s. *Geophysical Research Letters* 42, 104-112 (2015).
2. Kataoka T, Tozuka T, Behera S, Yamagata T. On the Ningaloo Niño/Niña. *Climate dynamics* 43, 1463-1482 (2014).
Kataoka, Takahito, et al. "On the ningaloo niño/niña." *Climate dynamics* 43 (2014): 1463-1482.
3. Doi T, Behera SK, Yamagata T. Predictability of the Ningaloo Niño/Niña. *Scientific Reports* 3, 2892 (2013).

6) Page 8; line 172: The convective centers change over P1 (IO) and P2 (WEP). Does that really mean an ‘eastward movement’ as suggested in the paper?

Response: Thank you for your comments. In the revised version of the manuscript, we adopted a more cautious approach, removing the direct conclusion regarding the "eastward movement" and instead described the convection center as being in different locations before and after 1985.

7) Page 11, and elsewhere in the paper: The physical explanations presented

in the paper suggest that the western Pacific warming (anthropogenic) can enhance an 'existing' decadal variability in the WAC – but doesn't induce a new mode of decadal variability. In that perspective, is it feasible to say an 'emergence of decadal linkage'? More appropriate would be to say, 'influence of western Pacific warming on the amplitude of decadal variability off the WAC'?

Response: Thank you for your suggestion. We denote "1985" as the "time of emergence" for the NNI-Niño4w linkage, defined as when the cross-spectrum of these two indexes is significant at 95% level, and their correlation coefficient is stable exceeding 95% significance. Here, the "emergence" we describe refers to the connection (correlation) between two regions reaching a certain threshold, rather than the creation of something new.

8) Fig. 4: Why does the trend estimation is limited to 1970 – 1995 for CMIP6 models? Is the phase of the IPO in CMIP models consistent with the observed IPO phase? If not necessarily (I think that is the case), how would the authors reject the fact that the linear relationship between the western Pacific SST and NNI amplitude might arise through IPO? I would suggest the authors add one more panel by redoing the same figure but once the IPO is regressed out from the western Pacific SST, to check this.

Response: We chose the period from 1970 to 1995 because we aim to investigate what leads to the sustained increase in the amplitude of the NNI during the positive IPO phase. We removed the influence of the IPO and calculated the correlation coefficient between the Western Pacific warming trend and the decadal NNI to be -0.03, the results were not statistically significant. We attribute the insignificant results to biases present within CMIP6 models. In the revised results, we have used pacemaker experiments related to the Indian Ocean. We found the important role of the rapid warming of the tropical Indian Ocean under external forcing in influencing the decadal variability of the two regions. Generally, Climate models (e.g., CMIP models, large ensemble simulations) do not adequately capture the phenomenon where the warming SST rate in the tropical Indian Ocean surpasses that in the tropical Pacific. Significant warming in the tropical Indian Ocean can induce negative sea-level pressure (SLP) anomalies that extend eastward to the western tropical Pacific, thereby intensifying the zonal SLP gradient across the Pacific and driving anomalous easterlies in the tropical western Pacific. However, in coupled models, weaker negative SLP anomalies are showed solely off western Australian coast (Zhang et al., 2019). This may explain why the results for the correlation with influence of IPO being removed were not as anticipated. The pacemaker experiments we added do not exhibit this bias, as they are capable of capturing the basin-wide negative SLP anomalies (Zhang et al., 2019). The results from pacemaker experiments related to Indian Ocean and large ensembles show that the Indian Ocean produces decadal fluctuations under the influence of external forcing since 1985, and being out of phase with the Pacific

and leading to an intensification of the easterly anomalies in the western-central tropical Pacific. Therefore, we no longer consider that a simple warming trend could cause decadal fluctuations in these two regions.

Reference:

1. Zhang L, et al. Indian Ocean warming trend reduces Pacific warming response to anthropogenic greenhouse gases: An interbasin thermostat mechanism. *Geophysical Research Letters* 46, 10882-10890 (2019).

9) Pacemaker experiment – The SST anomalies are nudged to the model climatology and HadISST1 anomalies. I would like to see the climatological SST map in the Indo-Pacific region from HadISST1 and the two models considered for the pacemaker experiment (left) and the spatial map of the standard deviation of the SST anomalies (for HadISST1 and two models) for the two periods (post and before 1980) in the right columns. And, please show the Niño4w & NN regions (box) over each of these maps.

Response: Thank you for your suggestion. For the climatological SST, the tropical Pacific SST in pacemaker experiments are warmer than the observation. For the standard deviation of the SST anomalies, the largest STD in pacemaker experiments occurs within the tropical Pacific region, indicating that the results of pacemaker experiment are predominantly driven by changes in the tropical Pacific.

Figure RB10 The climatological SST in **a** HadISST, **c** CESM1 Pacific pacemaker experiment, and **e** FGOALS-f3 Pacific pacemaker experiment before 1985. **b**, **d**, and **f** as a, c, and e, but after 1985.

Figure RB11 The standard deviation of the SST anomalies in **a** HadISST, **c** CESM1 Pacific pacemaker experiment, and **e** FGOALS-f3 Pacific pacemaker experiment before 1985. **b**, **d**, and **f** as a, c, and e, but after 1985.

Minor comments:

1) Previous studies indicate that half of the NN is NOT related to the western Pacific (La Nina), indicating an important local driving mechanism that controls wind and SST off the WAC (see Kataoka et al. 2013; Marshall et al. 2014). The respective role of local processes driving the enhanced decadal amplitudes is not well addressed in the present study. Can the authors include a few more notes about the anthropogenic effects on the local drivers, in the discussion?

Response: Thank you for your valuable comments. Indeed, we have primarily investigated the remote mechanisms influencing decadal timescale fluctuations of the Ningaloo Niño. However, local physical mechanisms are also very important for the occurrence and development of NN, especially the cyclonic atmospheric circulation anomalies off the northwest coast of Australia (Kataoka et al. 2014; Marshall et al. 2015). These anomalies reduce the prevailing southerly winds along the shore, leading to surface warming due to decreased latent heat loss and the increased poleward transport of warm tropical waters by the Leeuwin Current. This process is further facilitated by a local wind-evaporation-sea surface temperature (WES) feedback mechanism. On decadal timescale, the negative phase of IPO can induce positive sea surface height (SSH) anomalies in the equatorial western Pacific and the southeast Indian Ocean. At this time, there is a significant negative correlation between sea-level pressure (SLP) anomalies off the west coast of Australia and SSH

anomalies in the equatorial western Pacific and southeastern Indian Ocean on decadal timescale (Feng et al. 2015), this explains the role of the negative IPO. However, the warming in the Indian Ocean, which also induces basin-wide negative SLP anomalies (Zhang et al. 2019), is mainly controlled by external forcing, suggesting that external forcing impacts the local processes along the west coast of Australia.

We have added this part to the discussion:“ This study shows the enhanced decadal linkage and its impact on both regions. We have demonstrated the role of the Tropical Indian Ocean SSTA in this linkage, but in previous studies show that local processes off western Australian coast, such as cyclonic anomalies, are also important. On decadal scale, the negative IPO can induce positive sea surface height (SSH) anomalies in the equatorial western Pacific and the southeast Indian Ocean. Associated with this, there is a significant negative correlation between sea-level pressure (SLP) anomalies off western Australian coast and SSH anomalies in the equatorial western Pacific and southeastern Indian Ocean on decadal timescale (Feng et al. 2015). Another view, the rapid warming in the Indian Ocean, which can also induce basin-wide negative SLP anomalies (Zhang et al. 2019), is mainly controlled by increased anthropogenic forcings. This basin-wide SLP anomaly would induce cyclonic wind anomalies and impact the local processes off western Australian coast.”

Reference:

1. Kataoka T, Tozuka T, Behera S, Yamagata T. On the Ningaloo Niño/Niña. *Climate dynamics* 43, 1463-1482 (2014).
2. Marshall AG, Hendon HH, Feng M, Schiller A. Initiation and amplification of the Ningaloo Niño. *Climate Dynamics* 45, 2367-2385 (2015).
3. Feng M, et al. Decadal increase in Ningaloo Niño since the late 1990s. *Geophysical Research Letters* 42, 104-112 (2015).
4. Zhang L, et al. Indian Ocean warming trend reduces Pacific warming response to anthropogenic greenhouse gases: An interbasin thermostat mechanism. *Geophysical Research Letters* 46, 10882-10890 (2019).

2) Anthropogenic heat enters the ocean through natural variability (e.g., IPO). The global warming hiatus during the last decade is an example. It is hard to distinguish, hence, the decadal SST response to natural variability and external forcing over short periods. A coupled model experiment by Feng et al. (2015) indeed suggested that, even though external warming influences the preconditioning of the NN development, it (external forcing) itself may not cause the decadal modulation of the occurrence of NN over recent decades – I would ask authors to discuss these aspects and previous literature in the discussion of the paper.

Response: Thank you for your insightful comments. Restoring the sea temperatures in the eastern Pacific to observed value in experiments can

simulate significant negative SLP anomalies in the southeastern Indian Ocean. However, in a coupled model simulation with the same radiative forcing but without restoring the equatorial eastern Pacific SST, there are no significant negative sea level pressure anomalies in the southeast Indian Ocean (Feng et al. 2015). This precisely illustrates the importance we have emphasized on the rapid warming of the tropical Indian Ocean under external forcing for the decadal variability of the two regions. Climate models generally fail to capture the phenomenon that the warming rate of SST in the tropical Indian Ocean is higher than that of the tropical Pacific. When the tropical Indian Ocean warms significantly, it will cause negative SLP anomalies extending eastward to the western tropical Pacific, enhancing the zonal SLP gradient across the Pacific and driving the anomalous easterlies in the tropical western Pacific. In our pacemaker experiments related to Indian Ocean, the basin-wide negative SLP anomalies are reproduced (Zhang et al. 2019).

This extension of discussion is very necessary, and we have added some discussions in the revised manuscript: "When the tropical Indian Ocean experiences significant warming, it induces negative SLP anomalies extending eastward to the western tropical Pacific. This process amplifies the zonal SLP gradient across the Pacific, which, in turn, generates anomalous easterlies in the tropical western Pacific (Zhang et al. 2019). Nevertheless, a common shortfall among climate models is their inability to accurately represent this dynamic. Specifically, these models often underrepresent the discrepancy in warming rates between the tropical Indian Ocean and the tropical Pacific. The tropical Indian Ocean warms at a faster rate than the tropical Pacific, a critical factor that models fail to capture. This model bias results in an underestimation of the strong negative SLP anomalies. Consequently, this modeling bias could potentially influence the simulated SST off western Australian coast and in the western-central tropical Pacific."

Reference:

1. Feng M, et al. Decadal increase in Ningaloo Niño since the late 1990s. *Geophysical Research Letters* 42, 104-112 (2015).
2. Zhang L, et al. Indian Ocean warming trend reduces Pacific warming response to anthropogenic greenhouse gases: An interbasin thermostat mechanism. *Geophysical Research Letters* 46, 10882-10890 (2019).

3) Please provide the time series of decadal (>7 years) Niño4w and NNI together over 1950 – 2020 and provide the correlation coefficient for the periods 1950 – 2020, 1950 – 1985, and 1985 – 2020.

Response: Thank you for your good suggestion. We show the following figure, it is evident that the NNI and Niño4w index are out of phase, and their decadal variability enhanced after 1985.

Figure RB12 The timeseries for NNI and Niño4w with 7 years low-pass filtered. The blue line represents NNI, and the red line represents Niño4w index.

And we calculated their correlation coefficient on decadal timescale for the periods 1950 – 2020, 1950 – 1985, and 1985 – 2020 are -0.54, -0.63, and -0.68. For the raw timeseries, their correlation coefficient for the periods 1950 – 2020, 1950 – 1985, and 1985 – 2020 are -0.15, -0.34, and -0.49. The results show that their decadal linkage is stronger after 1985.

Reviewer #3 (Remarks to the Author):

Based on observations together with several sets of model simulations, i.e., pacemaker simulations from CESM1.2 and FGOALS-f3-L models, single-model CESM2-LE large ensemble simulations, and CMIP6 multi-model simulations, the authors investigated the coherence in decadal variations of sea surface temperature (SST) between the western Australian coast and the western-central Pacific. The coherence of SST variations on the decadal time scale between these two regions emerged in 1985. The authors claimed that the enhanced covarying decadal variations were attributed to the rapid warming in the western Pacific, however, the influence of Interdecadal Pacific Oscillation phase changes is relatively less prominent. The topic is important and interesting. The manuscript is in general well-organized and written, but some parts especially the methods could benefit from greater clarity. The conclusions that attribute the coherent decadal variations in the western Australian coast and the western-central Pacific to global warming merit further analyses/discussions to bring this forward more robustly.

Response: We appreciate the insightful comments from the review, which help to increase the reliability of our manuscript. According to the reviewer's suggestions, we have thoroughly revised the manuscript. We have provided a more detailed exposition of the method section, including the calculation of SST pattern formation, the criteria for selecting two groups of CESM2-LE members, and the calculation of effective degrees of freedom. We corrected the method of ensemble mean and added the display of significance test. We revised some results from the large ensemble and CMIP6 models, and in response to the bias

between the models and observations, we added two sets of pacemaker experiments related to the Indian Ocean and four single forcing large ensembles to explore the specific effects of warming and external forcing.

Major comments:

- The main conclusion and novelty of this study is attributing the decadal linkage between the western Australian coast and the western–central Pacific to the global warming induced by external forcing. This part of the results and discussion needs to be further enhanced. In the present manuscript, the authors use CESM2-LE large ensemble to investigate this and the ensemble mean of 100 members represents the forced signal. By comparing Fig. 3(c) and (d) based on CESM2-LE large ensemble mean to Fig. 1(d) and (e) based on observations, the decadal variations from the time scale of 8–16 years are quite different in the wavelet analysis results, especially the Niño4w index is mainly dominated by 3–7 year variations. The wavelet results in Fig. 3(c) and (d) don't have information of significance test as in Fig. 1 (d) and (e). This needs to be added and it will be helpful to add the time series of the ensemble mean NNI and Niño4w from CESM2-LE.

Response: Thank you for your valuable comments. In the revised version of our manuscript, we have taken into consideration numerous suggestions from you and recognize the importance of further discussing factors beyond the Interdecadal Pacific Oscillation (IPO). We provide the wavelet analysis of the NNI and Niño4w index from CESM2-LE. They exhibit enhanced decadal variabilities, but they do not pass the 95% significance test (Fig. RC1), the wavelet results are tested statistically against red noise (Torrence et al., 1998).

Instead, we supplemented our study with two sets of pacemaker experiments related to the Indian Ocean and four single forcing large ensembles to broaden our findings. In our revised manuscript, the wavelet analysis of the ensemble mean Niño4w index reveals decadal variability, and all wavelet analysis here includes statistical significance tests.

The wavelet analysis is applied to the ensemble mean NNI and Niño4w index from the CESM1 and FGOALS-f3-L Pacific pacemaker experiments (Fig. RC2; Fig. S2 of revised manuscript). The decadal signals in the POGA of CESM1 and that of FGOALS-f3-L are enhanced since the 1980s than those before the 1980s although the tests are not so significant at 95% level. The no-passing test is mainly because of the short time series. Beyond the IPO, there are other factors influencing the decadal variabilities and linkage between the two regions.

To further examine the role of Indian Ocean warming in the past decades, two pacemaker experiments related to Indian Ocean are also used: TIOGA using the National Center for Atmospheric Research (NCAR) Community Earth

System Model version 1 (CESM1), in which the simulated SST in the tropical Indian Ocean and part of the western Pacific warm pool region is restored to observed SST from Extended Reconstructed SST version 3b (ERSSTv3b), with fully restored SST anomalies (SSTA) for 15°N-15°S, from the African Coast to 161°E, and buffer zones at 15°N-20°N, 15°S-20°S, and 161°E-180°(Hurrell et al., 2013; Zhang et al., 2019; Yang et al., 2020). Another experiment is IOGA using NCAR CESM1, in which the simulated SST in the Indian Ocean (north of 50°S) is restored to observed SST from ERSSTv3b (Hurrell et al., 2013; Zhang and Han, 2021).

Both pacemaker experiments were able to demonstrate the enhancement of decadal variability in the two regions (Fig. RC3 b and f; Fig. S14 of revised manuscript). Results from TIOGA experiments reveal that the changes in tropical Indian Ocean and part of western Pacific can generate significant SST changes in the central-western tropical Pacific and off western Australian coast, accounting for a large fraction of the observed NNI after 1985; Before 1985, however, the simulated NNI has a weaker amplitude and opposes to the observation (Fig. RC4; Fig. 3c of revised manuscript). These results suggest that SSTA in tropical Indian Ocean and western Pacific play an important role in causing the enhanced NN-Niño4w linkage in recent decades. This is because a warm (cold) SSTA in tropical Indian-western Pacific basin drives equatorial easterly (westerly) wind anomaly in the central-western tropical Pacific (Fig. RC5b; Fig. S14 of revised manuscript), enhances (reduces) equatorial upwelling and causes cold (warm) SSTA in Niño4w region. The enhanced equatorial easterly (westerly) wind anomaly increases (reduces) the Indonesian Throughflow, strengthening Ningaloo Niño (Niña) events. Meanwhile, the colder (warmer) SSTA in Niño4w region can induce cyclonic (anticyclonic) circulation in the Southeast Indian Ocean, together with local air-sea feedback, further strengthening Ningaloo Niño (Niña) magnitudes.

In the TIOGA experiments, the NNI and Niño4w index consistently display negative correlation before and after 1985 (Fig. RC4d; Fig. 3d of revised manuscript), whereas the simulated NNI agrees with observed one only after 1985 (Fig. RC4c; Fig. 3c of revised manuscript). This is because before 1985, the Niño4w SSTA generated by Indian Ocean SST is out of phase with that of the IPO, whereas after 1985 they are generally in phase (Fig. RC6b; Fig. S15b of revised manuscript). The warm (cold) Indian Ocean SSTA and negative (positive) IPO act in concert to intensify Niño4w SSTA and thus NNI after 1985, while the Indian Ocean SSTA weakens the IPO effect before 1985 ((Fig. RB4f; Fig. 3 f of revised manuscript), causing the enhanced decadal linkage between NNI and Niño4w index after 1985 as shown by observations. Thus, the decadal linkage between NNI and Niño4w has been explained.

Before 1985, the warming of the tropical Indian Ocean was primarily influenced

by the Interdecadal Pacific Oscillation (IPO). Then the decadal variability in the Indian Ocean was in phase with the IPO (Dong et al., 2017; Zhang et al., 2018). After 1985, external forcing dominated the changes in the Indian Ocean (Fig. RC4e; Fig. 3e of revised manuscript). Under the influence of external forcing, the decadal variability in Indian Ocean and the IPO are out of phase (Fig. RC4f; Fig. 3f of revised manuscript). The warming of the Indian Ocean and the negative phase of IPO can both lead to enhanced easterlies in the Niño4w region (Fig. RC5b; Fig. S14b of revised manuscript).

Four single forcing large ensembles (Simpson et al., 2023) are employed to examine the role of external forcings further. They are GHG only (only greenhouse gases evolving, having 15 members); BMB only (Only biomass burning aerosols evolving, having 15 members); EE (everything else evolving, having 15 members, that is, all forcings are time evolving except GHGs, AAERs including BMB which means greenhouse gases and anthropogenic (including biomass burning) aerosols are held fixed at 1850 level; xAER (Everything time evolving except anthropogenic aerosols, having 10 members).

The above results show that, after 1985, external forcings caused TIO SST to be out of phase with the IPO, leading to decadal fluctuations. To further explore the role of external forcings, we compared the CESM2-LE and four single forcing large ensembles from CESM2. we showed that the increase in greenhouse gases (GHGs; Fig. RC7; Fig. 4 of revised manuscript, green line) has significantly contributed to the recent trend of rapid warming in the Tropical Indian Ocean. External forcings that exclude anthropogenic aerosols (experiment xAER; Fig. RC7; Fig. 4 of revised manuscript, blue line) produce a stronger warming trend, which means anthropogenic aerosols somewhat decrease the warming trend in the Indian Ocean. In addition to a warming trend, Experiment xAER also generates decadal variability in the tropical Indian Ocean SST Fig. RC7; (Fig. 4 of revised manuscript, blue line), mainly owing to the forcings by volcanoes (included in everything else experiment, EE; Fig. RC7; Fig. 4 of revised manuscript, purple line). This finding suggests that external forcings by volcanoes is the main cause for TIO SSTA and result in its out of phase relationship with IPO after 1985.

Figure RC1 SST variability off western Australian coast and the tropical western–central tropical Pacific in CSM2-LE. a Wavelet spectrum of the ensemble mean NNI for 1950–2014. The red lines mark the periodicity of 8–16 years. **b** Wavelet spectrum of the ensemble mean Niño4w index for 1950–2014. Stippled results are statistically significant at the 95% confidence level.

Figure RC2 SST variability off western Australian coast and the tropical western–central Pacific and their relationship in Pacific pacemaker experiments. a The 21-year sliding correlation coefficients between the ensemble mean NNI and the ensemble mean Niño4w index in CESM1.2 (pacemaker experiments), in which the red line signifies results statistically significant at the 95% confidence level. **b** Wavelet spectrum of the ensemble mean NNI for 1950–2014 in CESM1.2 (Pacific pacemaker experiments). The red lines mark the periodicity of 8–16 years. **c** Wavelet spectrum of the ensemble mean Niño4w index for 1950–2014 in CESM1.2 (Pacific pacemaker experiments). **d**, **e**, and **f** as in a, b, and c but in FGOALS-f3-L (pacemaker experiments) for 1950–2014.

Figure RC3 SST variability off western Australian coast and the tropical western–central tropical Pacific and their relationship in different Indian ocean pacemaker experiments. **a** The 21-year sliding correlation coefficients between the ensemble mean NNI and the ensemble mean Niño4w index in CESM1 tropical Indian ocean (TIOGA) pacemaker experiments, in which the red line signifies results statistically significant at the 95% confidence level. **b** Wavelet spectrum of the ensemble mean NNI for 1950–2013 in TIOGA. The red lines mark the periodicity of 8–16 years. **c** Wavelet spectrum of the ensemble mean Niño4w index for 1950–2013 in TIOGA. **d**, **e**, and **f** as in a, b, and c but in CESM1 Indian ocean (IOGA) pacemaker experiments for 1950–2019.

Figure RC4 Possible influencing factors (IPO and external forcing) for decadal variabilities of NNI. **a** Time series of the 21-year sliding variance of the NNI (black line) and the IPO index (blue line) based on ERSSTv5, in which

the red shading identifies the positive phase of the IPO during 1978–1997. **b** Time series of the 21-year sliding variance of interannual NNI (blue line) and decadal NNI (red line). Red shading identifies the positive phase of the IPO during 1978–1997. **c** The 8–16 years band-pass filtered decadal NNI in ERSSTv5 and TIOGA. **d** The 8–16 years band-pass filtered decadal NNI and Niño4w index in TIOGA. **e** The 8 years low-pass filtered tropical Indian ocean SSTA (15°N – 15°S , 50° – 160°E) in observations and the ensemble mean of CESM2-LE, and MPI-LE. Blue shading identifies the period 1960-1985. Red shading identifies the period after 1985. **f** The normalized 8-year low-pass filtered Tropical Indian ocean SSTA and IPO index (detrend) in observations.

Figure RC5 Climate teleconnections for the decadal SST variability off western Australian coast in DJF in different Indian Ocean pacemaker experiments. Correlation between the DJF decadal SSTA and the DJF decadal NNI (shading; vectors for wind anomaly; contours for precipitation anomaly) during **a** 1950–1990 and **b** 1990–2013 in CESM1 tropical Indian ocean (TIOGA) pacemaker experiments. **c** and **d** as in **a** and **b** but for the 200-hPa velocity potential anomaly (shading; vectors for divergent wind; contours for precipitation). Red/blue contours represent positive/negative precipitation significant at the 95% confidence level. Stippled areas are where the regression results are statistically significant at the 95% confidence level. **e**, **f**, **g**, and **h**

are for 1950–1990 and 1990–2019 in CESM1 Indian ocean (IOGA) pacemaker experiments.

Figure RC6 Decadal NNI and Niño4w index in IOGA and observation; IPO indexes in two large ensembles and different observation data. **a** The 8–16 years band-pass filtered decadal NNI in ERSSTv5 and IOGA. **b** The 8–16 years band-pass filtered decadal Niño4w index in ERSSTv5 and IOGA. **c** The 8 years low-pass filtered IPO index (15°N – 15°S , 50° – 160°E) in observations and the ensemble mean of CESM2-LE, and MPI-LE. Blue shading identifies the period 1960-1985. Red shading identifies the period after 1985.

Figure RC7 The contribution of single external forcing to the tropical Indian Ocean SSTA.

The 8 years low-pass filtered tropical Indian ocean SSTA (15°N – 15°S , 50° – 160°E) in ERSSTv5 (black) and the ensemble mean of CESM2-LE (red) and four sets of CESM2 single forcing large ensembles: xAER (blue), GHG (green), BMB

and EE (orange), and EE (purple). Light blue shading identifies the period 1960-1985. Light red shading identifies the period after 1985.

Reference:

1. Torrence C, Compo GP. A practical guide to wavelet analysis. *Bulletin of the American Meteorological society* 79, 61-78 (1998).
2. Hurrell JW, et al. The community earth system model: a framework for collaborative research. *Bulletin of the American Meteorological Society* 94, 1339-1360 (2013).
3. Zhang L, et al. Indian Ocean warming trend reduces Pacific warming response to anthropogenic greenhouse gases: An interbasin thermostat mechanism. *Geophysical Research Letters* 46, 10882-10890 (2019).
4. Yang D, et al. Role of tropical variability in driving decadal shifts in the Southern Hemisphere summertime eddy-driven jet. *Journal of Climate* 33, 5445-5463 (2020).
5. Zhang L, Han W. Indian ocean dipole leads to Atlantic Niño. *Nature Communications* 12, 5952 (2021).
6. Dong L, McPhaden MJ. Why has the relationship between Indian and Pacific Ocean decadal variability changed in recent decades? *Journal of climate* 30, 1971-1983 (2017).
7. Zhang L, Han W, Sienz F. Unraveling causes for the changing behavior of the tropical Indian Ocean in the past few decades. *Journal of Climate* 31, 2377-2388 (2018).
8. Simpson IR, et al. The CESM2 single-forcing large ensemble and comparison to CESM1: Implications for experimental design. *Journal of Climate* 36, 5687-5711 (2023).

- Fig. 4: What would the figure be like if the Y-axis was substituted with the correlation between NNI and Niño4w in the period before and after 1985? This would be more straightforward in supporting the enhanced decadal coherence between NNI and Niño4w due to external forcing-induced warming.

Response: Thank you for your good suggestion. Yes, the correlations between NNI and Niño4w across different periods would more directly demonstrate how the warming of the Western Pacific was strengthening the linkage between the two regions. In our previous manuscript, when we replaced the y-axis with the correlation between NNI and Niño4w for the periods before and after 1985, the results were not statistically significant. We attribute the insignificant results after modifying the y-axis to the simulated biases in the CMIP6 models.

In the revised results, we used pacemaker experiments related to the Indian Ocean. We found the important role of the rapid warming of the tropical Indian Ocean under external forcing in influencing the decadal variability of the two regions. Generally, Climate models (e.g., CMIP models, large ensemble simulations) do not adequately capture the phenomenon where the warming

SST rate in the tropical Indian Ocean surpasses that in the tropical Pacific. Significant warming in the tropical Indian Ocean can induce negative sea-level pressure (SLP) anomalies that extend eastward to the western tropical Pacific, thereby intensifying the zonal SLP gradient across the Pacific and driving anomalous easterlies in the tropical western Pacific. However, in coupled models, weaker negative SLP anomalies are showed solely off western Australian coast (Zhang et al., 2019). This may explain why the results for the correlation between NNI and Niño4w were not as anticipated. The pacemaker experiments we added do not exhibit this bias, as they are capable of capturing the basin-wide negative SLP anomalies (Zhang et al., 2019). The results from pacemaker experiments related to Indian Ocean and large ensembles show that the Indian Ocean produces decadal fluctuations under the influence of external forcing since 1985, and being out of phase with the Pacific and leading to easterly anomalies in the western–central Pacific. Therefore, we no longer consider that a simple warming trend could cause decadal fluctuations in these two regions.

Reference:

1. Zhang L, et al. Indian Ocean warming trend reduces Pacific warming response to anthropogenic greenhouse gases: An interbasin thermostat mechanism. *Geophysical Research Letters* 46, 10882-10890 (2019).

- The authors selected subsamples from CESM2-LE to investigate the influence of IPO on the enhanced covarying SST decadal variabilities in the western Australian coast and the western–central Pacific. It is unclear how the IPO phases of the selected members match observations. The Methods section explained shortly that the members are selected based on the IPO pattern correlation coefficient between the model and observation. How are the time-evolving IPO phases combined with the spatial patterns? I would suggest simply selecting the ensemble members that could resemble the IPO index phase changes from observations.

Response: Your thoughtful comment is appreciated. In the process of selecting CESM2-LE members with similar (R+) / opposite (R-) IPO phases, we prioritize the accurate simulation of IPO. Before engaging in time series selection, we calculate the pattern correlation coefficient between individual members and observed IPO to establish a baseline. We set a threshold of 0.5, ensuring that the selected members possess the capability to precisely simulate the IPO mode. Hence, the calculation of the correlation coefficient for spatial patterns is merely a preliminary step to find members that simulate the IPO mode. Following this step, we selected model members with IPO indices similar to and opposite from the observed IPO and obtained the ensemble mean results for each group.

Therefore, In the original manuscript, Fig. 3e presents the ensemble members

that could resemble the observed IPO index phase changes. However, taking into account your feedback and the model's inability to simulate the rate of SST warming in the Indian Ocean exceeding that of the Pacific (Zhang et al., 2019), we felt that relying solely on results from CESM2-LE is insufficient. Therefore, we have added two pacemaker experiments related to the Indian Ocean to replace the selecting CESM2-LE members with similar (R+) / opposite (R-) IPO phases.

Reference:

1. Zhang L, et al. Indian Ocean warming trend reduces Pacific warming response to anthropogenic greenhouse gases: An interbasin thermostat mechanism. *Geophysical Research Letters* 46, 10882-10890 (2019).

- L295-298 "Most climate models present relatively even warming in the tropical Pacific ...This model bias may affect the emergence timing of Indo-Pacific decadal linkage," What is the bias for CESM2? As CESM2-LE is crucial for this study, the warming pattern from this model should be presented.

Response: Thank you for your suggestion. We show the mean SST map and the SST bias between ERSSTv5 and CESM2-LE (CESM2-LE minus ERSSTv5). The warmer tropical Pacific Ocean in CESM2-LE indicate that it fails to capture observed La Niña-like cooling in the eastern Pacific (Fig. RC8).

Figure RC8 a The mean state of SST in CESM2-LE ensemble mean. **b** The

mean state of SST in ERSSTv5. **c** The bias between the CESM2-LE ensemble mean and ERSSTv5 (CESM2-LE ensemble mean minus ERSSTv5).

- It seems that the main analyses are on boreal winter time, i.e., December-January-February (DJF). In some places, it is stated, and in others not. It would be much clearer if the authors could state in general if seasonal or annual data was used in this study at the beginning of the manuscript and in the Method part. In case all the analyses are using DJF data, this information could be considered to be added to the abstract and even the title of the study.

Response: We appreciate the reviewer's valuable suggestions. In the first section of the results, the time series for NNI and Niño4w index include both raw data (Fig 1b and c, bars, original manuscript) and their decadal components (Fig 1b and c, black lines, original manuscript). Wavelet spectra analysis is based on the raw time series, while 21-years sliding correlations for both regions used annual time series, highlighting the broader context of decadal variations and connections between the two regions. In subsequent sections, considering warming events along Western Australia's coast occurring mainly during the boreal winter (Dec-Jan-Feb, DJF), we specifically focus on the boreal winter (DJF) for our study. We have emphasized this aspect for clarity in revised manuscript: "We used regression and correlation maps between seasonal mean variables (e.g., the SSTA and precipitation anomaly averaged for December-February) and the decadal NNI based on DJF mean to visualize the spatial pattern of the influence of the tropical Pacific on the region off western Australia, as previous studies have indicated that the Ningaloo Niño reaches its peak during boreal winter (Marshall et al, 2015; Kataoka et al, 2014; Feng et al, 2015; Zhang et al, 2018)."

Reference:

1. Marshall AG, Hendon HH, Feng M, Schiller A. Initiation and amplification of the Ningaloo Niño. *Climate Dynamics* 45, 2367-2385 (2015).
2. Kataoka T, Tozuka T, Behera S, Yamagata T. On the Ningaloo Niño/Niña. *Climate dynamics* 43, 1463-1482 (2014).
3. Feng M, et al. Decadal increase in Ningaloo Niño since the late 1990s. *Geophysical Research Letters* 42, 104-112 (2015).
4. Zhang L, Han W, Li Y, Shinoda T. Mechanisms for generation and development of the Ningaloo Niño. *Journal of Climate* 31, 9239-9259 (2018).

- The Methods are not explained very clearly.

i. The section "Analysis for the formation of SST patterns" is missing some information. The way leading from equation (1) to (2) is not well explained, and the Newtonian cooling component $Q_{lh}^{(o^{'})}$ is not found in equation (2).

Response: Thank you for this comment. We have added some details and

formulas in the revised version.

In the manuscript, we give the main conclusions and formulas, so Eq. (2) of original manuscript includes $Q_{lh}^{o'}$. Here is the detailed derivation process:

$$C \frac{\partial T'}{\partial t} = D'_0 + Q'_{net} \quad (1)$$

where the left-hand side of Eq. (1) equals zero; T' is the SST change; and Q'_{net} is the net surface heat flux anomaly ($Q'_{net} = Q'_{sw} + Q'_{lw} + Q'_{sh} + Q'_{lh}$; positive downward), which consists of shortwave radiation (Q'_{sw}), longwave radiation (Q'_{lw}), sensible heat flux (Q'_{sh}), and latent heat flux (Q'_{lh}).

On the decadal time scale, the MLT (temperature in the mixed layer) anomalies tendency term on the left-hand side of Eq. (1) is smaller than the two right-hand-side terms by at least one order of magnitude and can thus be neglected:

$$0 = D'_0 + Q'_{net} \quad (2)$$

Furthermore, the latent flux is decomposed into a Newtonian cooling $Q_{lh}^{o'}$ component and a residual that represents atmospheric forcing $Q_{lh}^{a'}$:

$$Q_{lh}^{o'} = \alpha \overline{Q_{lh}} T' \quad (3)$$

$$Q_{lh}^{a'} = Q'_{lh} - Q_{lh}^{o'} \quad (4)$$

Now, the net surface heat flux anomaly is:

$$Q'_{net} = Q'_{sw} + Q'_{lw} + Q'_{sh} + Q_{lh}^{o'} + Q_{lh}^{a'} \quad (5)$$

$$Q'_{net} = Q'_{sw} + Q'_{lw} + Q'_{sh} + Q_{lh}^{o'} + \alpha \overline{Q_{lh}} T' \quad (6)$$

Then, Eq. (2) is changed to:

$$-D'_0 = Q'_{sw} + Q'_{lw} + Q'_{sh} + Q_{lh}^{o'} + \alpha \overline{Q_{lh}} T' \quad (7)$$

and the equation of SST pattern can be written as follows:

$$T' = \frac{D'_0 + Q'_{sw} + Q'_{lw} + Q'_{sh} + Q_{lh}^{a'}}{\alpha \overline{Q_{lh}}} \quad (8)$$

Therefore, equation (8) includes $Q_{lh}^{o'}$.

ii. The amplitude of NNI on the decadal timescale is larger than that on the interannual timescale (refer to L95-96). Normally the interannual variability is more prominent with a larger amplitude than the decadal variability because the time series on the decadal time scale is more smoothed. I wonder how the amplitude is defined.

Response: Thank you for your feedback. The amplitude is defined as the standard deviations of time series at interannual and decadal timescales. In our analysis, the NNI was decomposed into seven different timescale components. The third and fourth components were combined as interannual signals covering 1–7 years, while the fifth and sixth components were combined as decadal signals with periods exceeding seven years. This decomposition allows us to differentiate and compare the amplitudes at interannual and decadal timescales.

iii. The IPO pattern correlation coefficient (PCC) is not introduced clearly how it is calculated. This is critical for understanding the results.

Response: Thank you for this comment. The pattern correlation coefficient (PCC) is the Pearson product-moment coefficient of linear correlation between two variables. In the process of selecting CESM2-LE members with similar (R+) / opposite (R-) IPO phases, we prioritize the accurate simulation of IPO. Before engaging in time series selection, we calculate the pattern correlation coefficient between individual members and observed IPO to establish a baseline. We set a threshold of 0.5, ensuring that the selected members possess the capability to precisely simulate the IPO mode. Therefore, we select the two groups of members based on the similarity of simulated and observed time series (IPO index).

- Please review and ensure the grammar is correct and keep the consistency in the use of either past or present tense throughout the manuscript.

Response: Thank you for constructive comments. As you suggested, we have revised the manuscript.

Minor comments:

- The first sentence in the abstract is incomplete or lacks information. Some words could be added to the end of the sentence to specify what impact the interbasin linkage would bring.

Response: We appreciate the reviewer's suggestion. To address the incomplete information in the first sentence of the abstract, we have included the following sentence: "The impact of interbasin linkage on the weather/climate and ecosystems is significantly broader and profounder than that of only appearing in an individual basin." This addition aims to specify the wide-ranging impacts of the teleconnection between the Indo-Pacific basins.

- L42-43: "...has experienced the most rapid warming..." I feel some conditions should be added, for instance "in the tropical regions", because the high latitude warming signal is stronger than the warming in these regions.

Response: We thank the reviewer for this suggestion. We have added "tropical"

in the sentence.

- L55: "...affect that in the Niño4w region..." What does the "that" refer to?

Response: We appreciate the reviewer's insightful comment. The use of "that" refers to the variability. To enhance clarity and conciseness, we have modified the sentence: "Conversely, the change off western Australian coast can actively affect variability in the Niño4w region, building a two-way interaction on interannual timescale" This revision aims to more precisely show the two-way interaction between the western Australian coast and the Niño4w region.

- L97: "in other areas around the region off western Australian coast," this is duplicated shown another time by the end of the sentence.

Response: Thank you for your correction. We have removed the repeated sentence.

- L135-136: "Note that the positive anomaly appears stronger in the southern South China Sea and the western Pacific", what does that mean?

Response: Thanks for this valuable comment. This sentence meant to show that the influence of Indo-Pac warm pool. To make this sentence more concise, we have changed it into "Note that positive SST anomalies appear in the western tropical Pacific west of the Niño4w region after 1985 while they are negligible before 1985."

-L148: "the norther wind..."-> "the northern wind..."

Response: We thank the reviewer for correction. We have modified this word. We also corrected all the similar grammar mistakes.

-L166: "the dynamic process has led to..." -> "the dynamic process via ocean heat transport has led to..."

Response: We thank the reviewer for correction. We have added it into this sentence.

-L173: "winter NNI and precipitation" -> It sounds a bit weird because the location of the NN region is in the southern hemisphere, the winter there is not in DJF. To keep consistent and be more precise, either stating the months or referring to the hemisphere would work.

Response: Thank you for this comments. We modified it into "boreal winter".

-L182-183: "...more in accord with their position in the climatological mean state..." Adding a figure of the climatological mean state will make it clearer.

Response: Thank you for your correction. Our intention was to indicate that during La Niña events, the Walker Circulation usually intensifies. This is due to the SST cooling in the central and eastern equatorial Pacific under La Niña conditions, leading to weakened atmospheric convection in that region, while

convection in the western–central Pacific is relatively strong. We should have corrected the relevant statements as “These changes make the ascending and descending motions associated with the NNI in the tropical region in accord with their position in the climatological La Niña state, thereby leading to the strengthening of the Walker circulation after 1985 under the La Niña”. However, in the revised manuscript, we have removed this sentence. Because we prefer to provide precise conclusions related to our results.

-L204: “Two sets of pacemaker experiments” Is it intended to say “with two models”?

Response: Thank you for your comment. Yes, two sets of pacemaker experiments include two models, i.e., CESM1.2 (with eight ensemble members) and FGOALS-f3-L (with ten ensemble members), respectively.

-L209: Why is the time period for CESM1.2 shorter than that for FGOALS-f3-L?

Response: Thank you for this comment. CESM1.2 is based on the CMIP5 historical forcings (~2005), and FGOALS-f3-L is based on CMIP6 forcings (~2014). In the revised manuscript, we have extended the CESM1.2 to 2014. The two sets of IPO pacemaker experiments have covering from 1950 to 2014 now.

-L222: “is similar to that observed exists for NNI...” -> “is similar to that observed for NNI...”

Actually, the decadal variability seems not similar between model simulations and observations, see my major comment.

Response: Thank you for this comment. In the original manuscript, the sentence “Out of 100 large ensemble members, the ensemble mean shows that decadal variability is similar to that observed exists for NNI (Fig. 3c) and Niño4w (Fig. 3d) in models.” Our purposes are to show that the SST variabilities of ensemble mean NNI and Niño4w. In original paper, we show the ensemble mean of wavelet. Now we show the wavelet analysis for ensemble mean NNI and Niño4w index with significance test. Both the NNI and Niño4w index exhibit enhanced decadal variabilities, yet the variabilities do not pass the 95% significance test (Fig. RC1). The above results also prompted us to explore the role of warming and external forcing further using models beyond CESM2-LE.

-L225-226: “The external forcing explains about 42.6% of the decadal variance of the NNI.” Where does this number come from? It is better to show the time series.

Response: Thank you for your valuable comment. We calculated the variance of NNI on decadal timescale in large ensemble and divided it by the variance of observed NNI on decadal timescale to obtain above numerical value.

We modified the method of ensemble mean and provided the time series of

ensemble means for NNI in CESM2-LE and MPI-LE, along with three sets of observational data, showing decadal fluctuations under the influence of external forcing after 1985 (Fig. RC9).

Figure RC9 The 8–16 years band-pass filtered NNI in observations and the ensemble mean of CESM2-LE, and MPI-LE. Blue shading identifies the period 1960-1985. Red shading identifies the period after 1985.

-L233-234: "The decadal linkage is consistent in the individual members due to large Signal-to-Noise." I don't understand this sentence.

Response: We appreciate your comments. The Signal-to-Noise Ratio (SNR) assesses the ratio of ensemble mean results (signal) to the standard deviation among all ensemble members (background noise). A higher SNR indicates a more robust capture of meaningful information across all ensemble members.

However, due to updates in our ensemble mean method and the pacemaker experiments used, the current results now employ the Student's t-test with effective degrees of freedom:

$$\frac{1}{N_{eff}} \approx \frac{1}{N} + \frac{2}{N} \sum_{j=1}^{N-2} \frac{N-j}{N} \rho_{XX}(j) \rho_{YY}(j)$$

where N is the sample size, $\rho_{XX}(j)$ and $\rho_{YY}(j)$ are the autocorrelations of two time series X and Y at lag.

Reference:

1. Li J, et al. Influence of the NAO on wintertime surface air temperature over East Asia: multidecadal variability and decadal prediction. *Advances in Atmospheric Sciences* 39, 625-642 (2022).

-L245: "convection in situ"-> "in situ convection"

Response: We thank the reviewer for correction. We have modified this sentence.

-L250: "correlation coefficient (0.6) between the decadal NNI and the warming trend", what would the value of correlation coefficient be when the decadal NNI is substituted with the correlation between decadal NNI and Niño4w?

Response: Thank you for your good suggestion. Yes, the correlations between

NNI and Niño4w across different periods would more directly demonstrate how the warming of the Western Pacific was strengthening the linkage between the two regions. We replaced the y-axis with the correlation between NNI and Niño4w for the periods before and after 1985. Before 1985, the value is -0.11, and after 1985, the value is -0.26. Although the results indicated that the increased western Pacific warming led to an increase in the negative correlation between the two regions, the results did not pass the 95% significance test.

We attribute the insignificant results after modifying the y-axis to the simulated biases in the CMIP6 models. In the revised results, we have used pacemaker experiments related to the Indian Ocean. We found the important role of the rapid warming of the tropical Indian Ocean under external forcing in influencing the decadal variability of the two regions. Generally, Climate models (e.g., CMIP models, large ensemble simulations) do not adequately capture the phenomenon where the rate of SST warming in the tropical Indian Ocean surpasses that in the tropical Pacific. Significant warming in the tropical Indian Ocean can induce negative sea-level pressure (SLP) anomalies that extend eastward to the western tropical Pacific, thereby intensifying the zonal SLP gradient across the Pacific and driving anomalous easterlies in the tropical western Pacific. However, in coupled models, weaker negative SLP anomalies are showed solely off western Australian coast (Zhang et al., 2019). This may explain why the results for the correlation between NNI and Niño4w were not as anticipated. The pacemaker experiments we added do not exhibit this bias, as they are capable of capturing the basin-wide negative SLP anomalies (Zhang et al., 2019). The results from pacemaker experiments related to Indian Ocean and large ensembles show that the Indian Ocean produces decadal fluctuations under the influence of external forcing since 1985, and being out of phase with the Pacific and leading to an intensification of the easterly anomalies in the western–central tropical Pacific. Therefore, we no longer consider that a simple warming trend could cause decadal fluctuations in these two regions.

Reference:

1. Zhang L, et al. Indian Ocean warming trend reduces Pacific warming response to anthropogenic greenhouse gases: An interbasin thermostat mechanism. *Geophysical Research Letters* 46, 10882-10890 (2019).

-L303: "by Feng et al.²⁴ and Trenery and Han²⁶." Why not just add the numbers as citations?

Response: Thank you, we have removed them to make the sentence more concise.

-L308: "It is unnecessary to cite the paper here again. It reads like reference #37 is a previous publication of the authors. BTW, the information on this paper in the reference list is incomplete."

Response: Thank you, we have removed the reference #37 of original manuscript and we reedited the information of the reference.

-L345: "with eight simulations" do you mean "ensemble members"?

Response: Thank you. Yes, it means that CESM1.2-pacemaker includes eight ensemble members. In revised manuscript, we have corrected it to "with eight members".

-L346: "31 simulations" models? Simulations from 31 Earth system models?

Response: Thank you for the correction, it should be "31 climate models" from CMIP6.

L359: "1000 EEMD ensemble members ...0.2. ..." Is 1000 a typical number of members? What does the magnitude of standard deviation of 0.2 mean to the results? How was the number determined, would a smaller value of 0.1 or a bigger value of 0.3 work?

Response: Yes, 1000 is a typical number for the ensemble size based on previous studies. A standard deviation magnitude of 0.2 for added noise is common. In large ensemble means, the added noise theoretically cancels each other out, making the differences in results with various noise amplitudes minor. According to Qian (2016), increasing the ensemble size compensates for larger noise amplitudes. We have added the reference in revised manuscript.

Reference:

1. Qian C. Disentangling the urbanization effect, multi - decadal variability, and secular trend in temperature in eastern China during 1909–2010. Atmospheric Science Letters 17, 177-182 (2016).

L366: "by dividing each monthly anomaly by the standard deviation"-> "by dividing each monthly anomaly with the standard deviation"

Response: Thank you for the correction, we have replaced "by" with "with".

L369: to have the full name of WES

Response: Thank you, we have added the full name of WES feedback here. WES feedback is wind–evaporation–SST feedback.

L370: the citation should be #29, the #32 is not Xie et al.

Response: Thank you, we have corrected the citation.

L377: $Q_{lh}^{o'}$ This term does not appear in the equation (2)

Response: Thank you for this comment. The latent flux is decomposed into a Newtonian cooling $Q_{lh}^{o'}$ component and a residual that represents atmospheric

forcing $Q_{lh}^{\alpha'}$:

$$Q_{lh}^{\alpha'} = \alpha \overline{Q_{lh}} T' \quad (1)$$

$$Q_{lh}^{\alpha'} = Q'_{lh} - Q_{lh}^{\alpha'} \quad (2)$$

and the equation of SST pattern can be written as follows:

$$T' = \frac{D'_0 + Q'_{sw} + Q'_{lw} + Q'_{sh} + Q_{lh}^{\alpha'}}{\alpha \overline{Q_{lh}}} \quad (3)$$

Therefore, equation (3) includes $Q_{lh}^{\alpha'}$.

L381: "where Q'_{lw} and Q'_{sh} are neglected to owe to their.." This is confusing. These two terms are presented in Table S2. Is this statement based on the results there? Then it should be in the main text. This sentence is not grammatically correct.

Response: In previous study, Q'_{lh} plays an important role of SST pattern formation (Huang, et al., 2019; Xie, et al., 2010). For the Q'_{net} , it is usually dominant by the Q'_{sw} and $Q_{lh}^{\alpha'}$. In our manuscript, we show the similar results too. In order to avoid ambiguity, we have removed this sentence from the methods section.

Reference:

1. Huang Y, Wu B, Li T, Zhou T, Liu B. Interdecadal Indian Ocean basin mode driven by interdecadal Pacific oscillation: A season-dependent growth mechanism. *Journal of Climate* 32, 2057-2073 (2019).
2. Xie S-P, Deser C, Vecchi GA, Ma J, Teng H, Wittenberg AT. Global warming pattern formation: Sea surface temperature and rainfall. *Journal of Climate* 23, 966-986 (2010).

L390: decadal NNI Is NNI also based on DJF mean or annual mean data?

Response: Thank you for the comments. In the time series, we present the results of filtering the original monthly indices over 8–16 years; whereas, in the regression or correlation map results, we use the decadal NNI based on DJF mean. In the revised version, we have added the description of this part in the methods: "We used regression and correlation maps between seasonal mean variables (e.g., the SSTA and precipitation anomaly averaged for December-February) and the decadal NNI based on DJF mean to visualize the spatial pattern of the influence of the tropical Pacific on the region off western Australia, as previous studies have indicated that the Ningaloo Niño reaches its peak during boreal winter (Marshall et al, 2015; Kataoka et al,2014; Feng et al, 2015; Zhang et al, 2018)."

Reference:

1. Marshall AG, Hendon HH, Feng M, Schiller A. Initiation and amplification of the Ningaloo Niño. *Climate Dynamics* 45, 2367-2385 (2015).
2. Kataoka T, Tozuka T, Behera S, Yamagata T. On the Ningaloo Niño/Niña. *Climate dynamics* 43, 1463-1482 (2014).
3. Feng M, et al. Decadal increase in Ningaloo Niño since the late 1990s. *Geophysical Research Letters* 42, 104-112 (2015).
4. Zhang L, Han W, Li Y, Shinoda T. Mechanisms for generation and development of the Ningaloo Niño. *Journal of Climate* 31, 9239-9259 (2018).

L392-393: From observation, we choose the CESM2-LE members with IPO phases similar (R+) /opposite (R-). I don't understand this. Maybe the authors meant "According to" or "Based on" observation? "with IPO phases similar (R+) /opposite (R-)." -> "with similar (R+) /opposite (R-) IPO phases."

Response: Thank you for the suggestion. Yes, we meant "based on" observation. The sentence "we choose the CESM2-LE members with similar (R+) /opposite (R-) IPO phases" is more precise. We have revised the methods; thus, this content is no longer used.

L393-394: "Before choosing" could be removed.

Response: Thank you for this comment. We have removed this sentence.

L394-395: "PCC" is only used here and in the next sentence, I would suggest not introducing a new abbreviation.

Response: Thank you for this suggestion. We have replaced the abbreviation PCC with the pattern correlation coefficient. In revised manuscript, the related content is no longer used.

Fig. 1: There is a typo in the title of Fig. 1(g): "Corss" -> "Cross".

Response: Thank you for this correction. We have revised the Fig. 1(g).

Fig. 2: Regarding the decadal NNI, there are only 3-4 cycles within 35 years, what is the degree of freedom for the statistically significant test?

Response: Thank you for your valuable comments. We employed a two-tailed Student's t-test to assess statistical significance. Given our application of low-pass filtering and similar methods, which reduces degrees of freedom, we utilize effective degrees of freedom in our calculations. The formula used is:

$$\frac{1}{N_{eff}} \approx \frac{1}{N} + \frac{2}{N} \sum_{j=1}^{N-2} \frac{N-j}{N} \rho_{XX}(j) \rho_{YY}(j)$$

where N is the sample size, $\rho_{XX}(j)$ and $\rho_{YY}(j)$ are the autocorrelations of two time series X and Y at lag. The details have been added in revised manuscript.

Reference:

1. Li J, et al. Influence of the NAO on wintertime surface air temperature over East Asia: multidecadal variability and decadal prediction. *Advances in Atmospheric Sciences* 39, 625-642 (2022).

The precipitation contours in Fig. 2(c)-(d) are hard to find and the values are missing.

Response: Thank you for this valuable comments. We have redrawn the regression maps for precipitation (Fig. RC10; Fig. S7 of revised manuscript), and the current Fig. 2(c)-(d) now show the 200hPa velocity potential and divergent winds more clearly (Fig. RC11; Fig. 2 of revised manuscript) in revised manuscript. Precipitation is displayed separately in a single figure (Fig. RC10; Fig. S7 of revised manuscript).

Figure RC10 The increased precipitation over the Maritime Continent. Regression of the DJF precipitation onto the DJF decadal NNI in a 1950–1985 and b 1985–2020 for ERSSTv5. Stippled areas are where the regression results are statistically significant at the 95% confidence level. Panels c and d are based on HadISST1, and panels e and f are based on IAP data.

Figure RC11 Teleconnections for the decadal SST variability off western Australian coast in December–January–February (DJF). Regression of the DJF SSTA onto the DJF decadal NNI (shading; unit: K; vectors for wind anomaly; unit: m/s) in a 1950–1985 and b 1985–2020. The blue/red box represents the region off western Australian coast and the Niño4w region, respectively. Stippled areas are where the regression results are statistically significant at the 95% confidence level. c and d as in a and b but for the 200-hPa velocity potential anomaly (shading; unit: 106 m/s; vectors for divergent wind; unit: m/s;). Ningaloo Niño generally peaks in boreal winter; thus, only the changes in boreal winter are presented.

“Ningaloo Niño generally peaks in boreal winter; thus, only the changes in boreal winter are presented.” -Is this a general statement that could be applied to the whole study and all the figures shown in the manuscript? In that case, this needs to be mentioned at the beginning of the manuscript and introduced in the methods.

Response: Thank you for this good suggestion. For the wavelet analysis, we present the results based on the original data. For the time series, we provide results from either the original data or the original data subjected to 8–16 years band-pass filtering. For the spatial maps of regression or correlation, we use the decadal NNI based on the DJF mean, as previous studies have indicated that the Ningaloo Niño reaches its peak during boreal winter (Kataoka et al. 2014; Feng et al. 2015; Marshall et al. 2015). We have explained this in a straightforward manner in the methods section of our revised manuscript: “We used regression and correlation maps between seasonal mean variables (e.g., the SSTA and precipitation anomaly averaged for December–February) and the decadal NNI based on DJF mean to visualize the spatial pattern of the influence of the tropical Pacific on the region off western Australia, as previous studies have indicated that the Ningaloo Niño reaches its peak during boreal winter (Marshall et al, 2015; Kataoka et al, 2014; Feng et al, 2015; Zhang et al, 2018).”

Reference:

1. Marshall AG, Hendon HH, Feng M, Schiller A. Initiation and amplification of the Ningaloo Niño. *Climate Dynamics* 45, 2367-2385 (2015).
2. Kataoka T, Tozuka T, Behera S, Yamagata T. On the Ningaloo Niño/Niña. *Climate dynamics* 43, 1463-1482 (2014).
3. Feng M, et al. Decadal increase in Ningaloo Niño since the late 1990s. *Geophysical Research Letters* 42, 104-112 (2015).
4. Zhang L, Han W, Li Y, Shinoda T. Mechanisms for generation and development of the Ningaloo Niño. *Journal of Climate* 31, 9239-9259 (2018).

Fig. 3: The way of choosing the R+/R- group in this study might be affected by the model's response of SSTA to IPO. It's possible that some SSTA patterns may not be related to certain IPO phases. As the purpose is to check the IPO, it is more straightforward to choose the ensemble members by the simulated IPO indices. Would this way end up with two similar sets of ensemble members?
Response: Thank you for this suggestion. Calculating the spatial correlation coefficient was merely to ensure that the model's members could simulate the IPO mode, rather than directly selecting members based on their spatial correlation with observations. As you pointed out, we selected members based on their similarity to or opposition against the observed IPO index. However, we also recognized that this selection method might not effectively distinguish the influence of the IPO. Therefore, we utilized two pacemaker experiments related to the Indian Ocean and four single forcing large ensembles to further investigate.

Fig. 4: I would suggest adding figures with substituting the y-axis with correlation coefficients between NNI and Niño4w before and after 1985.
Response: Thank you for your good suggestion. Yes, the correlations between NNI and Niño4w across different periods would more directly demonstrate how the warming of the Western Pacific was strengthening the linkage between the two regions. We replaced the y-axis with the correlation between NNI and Niño4w for the periods before and after 1985. Before 1985, the value is -0.11, and after 1985, the value is -0.26. Although the results indicated that the increased western Pacific warming led to an increase in the negative correlation between the two regions, the results did not pass the 95% significance test.

We attribute the insignificant results after modifying the y-axis to the simulated biases in the CMIP6 models. In the revised results, we have used pacemaker experiments related to the Indian Ocean. We found the important role of the rapid warming of the tropical Indian Ocean under external forcing in influencing the decadal variability of the two regions. Generally, Climate models (e.g., CMIP models, large ensemble simulations) do not adequately capture the phenomenon where the warming SST rate in the tropical Indian Ocean surpasses that in the tropical Pacific. Significant warming in the tropical Indian Ocean can induce negative sea-level pressure (SLP) anomalies that extend

eastward to the western tropical Pacific, thereby intensifying the zonal SLP gradient across the Pacific and driving anomalous easterlies in the tropical western Pacific. However, in coupled models, weaker negative SLP anomalies are showed solely off western Australian coast (Zhang et al., 2019). This may explain why the results for the correlation between NNI and Niño4w were not as anticipated. The pacemaker experiments we added do not exhibit this bias, as they are capable of capturing the basin-wide negative SLP anomalies (Zhang et al., 2019). The results from pacemaker experiments related to Indian Ocean and large ensembles show that the Indian Ocean produces decadal fluctuations under the influence of external forcing since 1985, and being out of phase with the Pacific and leading to an intensification of the easterly anomalies in the western–central tropical Pacific. Therefore, we no longer consider that a simple warming trend could cause decadal fluctuations in these two regions.

Reference:

1. Zhang L, et al. Indian Ocean warming trend reduces Pacific warming response to anthropogenic greenhouse gases: An interbasin thermostat mechanism. *Geophysical Research Letters* 46, 10882-10890 (2019).

Table S2: It will be easier to add the long names of the terms into the table after each term.

Response: Thank you for suggestion. We have revised the Table S2 to make it concise.

Table RC1 Contribution of WES feedback and oceanic heat transport to the formation of SST pattern. Regression of the DJF SSTA (T' , unit: K; Eq. (2)) onto the DJF decadal NNI (out of brackets) and DJF NNI (in brackets) for 1958–1985 and 1985–201). Also shown are the contributions of latent heat flux to the atmosphere (Q'_{lk} , unit: K), oceanic heat transport (D'_0 , unit: K), shortwave radiation (Q'_{sw} , unit: K), longwave radiation (Q'_{lw} , unit: K), and sensible heat flux (Q'_{sh} , unit: K).

	Before 1985	After1985
T'	0.12 (0.88)	0.4 (0.92)
Q'_{lk}	0.32 (1.21)	0.33 (1.29)
D'_0	-0.25 (-0.43)	0.15 (0.004)
Q'_{sw}	-0.05 (0.04)	-0.14 (-0.48)
Q'_{lw}	0.03 (-0.02)	0.01 (0.03)

Q'_{sh}

0.07 (0.08)

0.05 (0.08)

REVIEWERS' COMMENTS

Reviewer #3 (Remarks to the Author):

The authors have done lots of additional analyses and clarified further the methods. In general, my concerns are mostly addressed in the revised manuscript and responses by the authors. I have only minor comments for clarification.

- Why are the new results of wavelet spectrum of SSTA in NNI and Nino4w from CESM2-LE in Figure RC1 so different from the previous results of wavelet spectrum of NNI and Nino4w indices shown in Fig. 3 (c)-(d) in the original submitted manuscript?

- Table S1. It would be more straightforward to have the full name of each term in the table rather than in the caption. There are enough spaces in the table. The words in the caption are unclear and should be revised, this is an example: “onto the DJF decadal NNI (out of brackets) and DJF NNI (in brackets)” -> “onto the DJF NNI on decadal timescale and the DJF NNI of original time series without filtering (in brackets)”

Reviewer #3 (Remarks to the Author):

The authors have done lots of additional analyses and clarified further the methods. In general, my concerns are mostly addressed in the revised manuscript and responses by the authors. I have only minor comments for clarification.

Response: We appreciate the insightful comments from the review, which help to increase the reliability of our manuscript. According to the reviewer's comments, we have further clarified the reasons for the slight differences between Fig. 3 (c)-(d) in the original and revised manuscripts, and revised Table S1 based on the reviewer's suggestions.

- Why are the new results of wavelet spectrum of SSTA in NNI and Nino4w from CESM2-LE in Figure RC1 so different from the previous results of wavelet spectrum of NNI and Nino4w indices shown in Fig. 3 (c)-(d) in the original submitted manuscript?

Response: We thank the reviewer for such an insightful comment. We initially calculated the wavelet for each ensemble member and then average them in original submitted manuscript. This method will cause the results to be affected by internal variability.

In revised manuscript, we focus on studying the role of external forcing in revised manuscript. To ensure that the presented results are entirely generated by external forcing, we have corrected the analysis processes and update the results. Here, all related results are first subjected to the ensemble mean from the related members, then further process (including wavelet analysis) based on the ensemble mean.

- Table S1. It would be more straightforward to have the full name of each term in the table rather than in the caption. There are enough spaces in the table. The words in the caption are unclear and should be revised, this is an example: "onto the DJF decadal NNI (out of brackets) and DJF NNI (in brackets)" -> "onto the DJF NNI on decadal timescale and the DJF NNI of original time series without filtering (in brackets)"

Response: Thank you for the reviewer's valuable suggestions. We have revised the text according to the suggestions and added the full names of each item to the table to make it more comprehensible.

Table S1. Contribution of WES feedback and oceanic heat transport to the formation of SST pattern. Regression of the DJF SSTA (T' , unit: K; Eq. (2)) onto the DJF NNI on decadal timescale and the DJF NNI of original time series without filtering (in brackets) for 1958–1985 and 1985–2015). Also shown are the contributions of latent heat flux to the

atmosphere ($Q_{lh}^{a'}$, unit: K), oceanic heat transport (D'_0 , unit: K), shortwave radiation (Q'_{sw} , unit: K), longwave radiation (Q'_{lw} , unit: K), and sensible heat flux (Q'_{sh} , unit: K).

	Before 1985	After1985
SST (T')	0.12 (0.88)	0.4 (0.92)
Latent heat flux to the atmosphere ($Q_{lh}^{a'}$)	0.32 (1.21)	0.33 (1.29)
Oceanic heat transport (D'_0)	-0.25 (-0.43)	0.15 (0.004)
Shortwave radiation (Q'_{sw})	-0.05 (0.04)	-0.14 (-0.48)
Longwave radiation (Q'_{lw})	0.03 (-0.02)	0.01 (0.03)
Sensible heat flux (Q'_{sh})	0.07 (0.08)	0.05 (0.08)